# Interpolation and differentiation of alchemical degrees of freedom in machine learning interatomic potentials

Juno Nam ⬡, Jiayu Peng ⬡ & Rafael Gómez-Bombarelli ⬡ ✉

Machine learning interatomic potentials (MLIPs) have become a workhorse of modern atomistic simulations, and recently published universal MLIPs, pre-trained on large datasets, have demonstrated remarkable accuracy and generalizability. However, the computational cost of MLIPs limits their applicability to chemically disordered systems requiring large simulation cells or to sample-intensive statistical methods. Here, we report the use of continuous and differentiable alchemical degrees of freedom in atomistic materials simulations, exploiting the fact that graph neural network MLIPs represent discrete elements as real-valued tensors. The proposed method introduces alchemical atoms with corresponding weights into the input graph, alongside modifications to the message-passing and readout mechanisms of MLIPs, and allows smooth interpolation between the compositional states of materials. The end-to-end differentiability of MLIPs enables efficient calculation of the gradient of energy with respect to the compositional weights. With this modification, we propose methodologies for optimizing the composition of solid solutions towards target macroscopic properties, characterizing order and disorder in multicomponent oxides, and conducting alchemical free energy simulations to quantify the free energy of vacancy formation and composition changes.

Atomistic simulations are a cornerstone of computational modeling of the dynamic behavior of materials. Achieving predictive and efficient simulations necessitates a balance between the quality and cost of the description of interatomic interactions and exhaustive sampling to achieve converged thermodynamic averages. Density functional theory (DFT) calculations are typically taken as a gold standard for accuracy in materials simulations. Ab initio molecular dynamics (AIMD) simulations[1] propagate dynamics using these high-quality DFT forces, but their high computational cost limits scalability. Machine learning interatomic potentials (MLIPs)[2,3], trained on electronic structure calculation results, offer a low-cost alternative to DFT energies and forces in MD. Beginning from the seminal works of the Behler–Parrinello network[4] and GAP[5], various architectures of MLIP have been proposed to offer a selection within a trade-off between accuracy and speed,

such as SchNet[6], PaiNN[7], NequIP[8], Allegro[9], MACE[10,11], and CACE[12]. Recently, universal MLIPs, such as M3GNet[13], CHGNet[14], and MACE-MP-0[15], have emerged, providing atomistic modeling capabilities across a substantial portion of elements in the periodic table and their combinations. All these models are trained on DFT energies and gradients extracted from a large-scale materials database such as the Materials Project[16]. The benchmark results[17,18] demonstrate that they offer high-fidelity modeling of atomic interactions and phonon dispersion, thereby serving as reliable foundation models in the context of downstream atomistic simulation applications.

While interatomic potentials are primarily intended to operate on atomic positions with fixed elemental identities, it is intriguing to consider their alchemical degrees of freedom, wherein the elemental identities can be altered continuously. In the realm of electronic

Department of Materials Science and Engineering, Massachusetts Institute of Technology, Cambridge, MA, USA. ✉e-mail: rafagb@mit.edu

structure methods, von Lilienfeld and colleagues have pioneered the molecular grand-canonical ensemble DFT and have advanced subsequent lines of research on alchemical transformations, which enable the alteration and optimization of chemical compositions[19–23]. From the standpoint of MLIPs, Ceriotti and colleagues introduced an alchemical compression scheme based on an atom-centered density framework and applied the approach to model high-entropy alloys[24–26]. They demonstrated that compressing the representation of physical elements onto low-dimensional subspaces of pseudoelements enables efficient modeling of compositionally complex systems and interpolation to elements not encountered during training. Additionally, Chen et al.[27] demonstrated that pre-trained materials property predictors can be applied to disordered crystals by using linear interpolation of low-dimensional elemental embeddings. While continuous representations of elements correspond to atomic embeddings in graph-based MLIPs, most universal MLIPs typically use much higher-dimensional atomic embeddings to ensure that the model is sufficiently expressive. Since models are only trained with discrete atom identities, it is challenging to identify meaningful submanifolds of elemental embeddings to interpolate elements or project gradients, as seen in the context of molecule design with pre-trained MLIPs[28]. On the other hand, simple linear interpolation of embeddings for modeling compositions may lead to unphysical outputs.

Alchemical changes are of particular importance in free energy simulations[29,30]. Free energy simulations are widely used to characterize the finite-temperature stabilities of solid phases[31,32], and automatic protocols have been developed accordingly[33]. However, while alchemical free energy calculations are widely used to study protein–small molecule interactions[34], their applications in materials systems are limited. This would be largely due to the challenge of parametrizing interatomic potentials for systems with three or more elements. Notably, Jinnouchi et al.[35] introduced a thermodynamic integration (TI) method to compute the chemical potentials of liquid Si and LiF in $H_2O$ by smoothly turning on or off interactions between atoms in kernel-based MLIPs through alchemical switching.

With the advent of universal MLIPs, the challenge of fitting potentials for systems containing multiple types of elements has been alleviated, and they provide reasonable accuracy for dynamics around equilibrium geometries. Thus, it is timely to consider the application of universal MLIPs to facilitate free energy simulations along alchemical pathways. In this work, building upon the prototypical construction of graph-based MLIPs, we access the hitherto hidden alchemical degrees of freedom inherent in MLIPs. Rather than altering the continuous embeddings of individual atoms, we augment the input graph

structure by introducing alchemical atoms, each associated with its respective compositional weight. Through subsequent modifications to the message passing scheme and energy readout, our scheme provides smooth interpolation between different compositional states of materials. Moreover, given the end-to-end differentiability with respect to the alchemical weights $\boldsymbol{\lambda}$, it facilitates the calculation of the alchemical gradient of the energy $\partial H/\partial \boldsymbol{\lambda}$ and subsequently the calculation of the free energy of the alchemical transformation. In addition, we explore the application of alchemical intermediate states with mixed compositions in creating a computationally efficient description of solid solutions.

## Results

### Alchemical graph and message passing

**Prototypical MLIP construction.** Our objective here is to introduce modifications to the non-learnable parts of the MLIPs so that we can model the alchemical compositions of materials without further fine-tuning the models. First, we start by introducing the prototypical construction of graph-based MLIPs. An atomic system is represented as a graph $\mathcal{G} = (\mathcal{V}, \mathcal{E})$ with an atom as a node $i \in \mathcal{V}$ and an atom pair within a defined cutoff distance as an edge $(i,j) \in \mathcal{E}$[36,37]. Each element $Z_i$ is embedded into a continuous vector $\boldsymbol{z}_i$, which is then used to initialize node features $\boldsymbol{h}_i^{(0)}$. Edge features $\boldsymbol{e}_{ij}$ are derived from the relative displacements $\boldsymbol{r}_{ij}$. The input is then passed through the layers of the graph neural network with a message-passing mechanism[38–40]. In layer $t$, a message $\boldsymbol{m}_i^{(t)}$ is constructed by pooling the message contributions over the neighboring nodes $\mathcal{N}(i)$ as

$$\boldsymbol{m}_i^{(t)} = \sum_{j \in \mathcal{N}(i)} M_t\left(\boldsymbol{h}_i^{(t)}, \boldsymbol{h}_j^{(t)}, \boldsymbol{e}_{ij}\right),\qquad(1)$$

where each contribution is computed from the hidden node features and the edge feature by a message function $M_t$. The messages are then used to update the node features:

$$\boldsymbol{h}_i^{(t+1)} = U_t\left(\boldsymbol{h}_i^{(t)}, \boldsymbol{m}_i^{(t)}\right),\qquad(2)$$

where $U_t$ is an update function. Finally, a readout function $R$ transforms the final node features $\boldsymbol{h}_i^{(T)}$ into the node energies, which are summed over the entire node list to give an estimate of the potential energy as

$$E = \sum_{i \in \mathcal{V}} R\left(\boldsymbol{h}_i^{(T)}\right).\qquad(3)$$

**a** Graph augmentation    **b** Message passing    **c** Energy readout

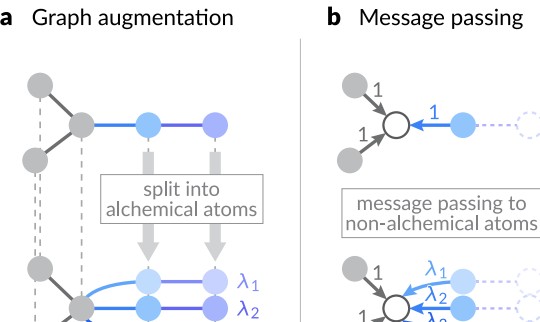
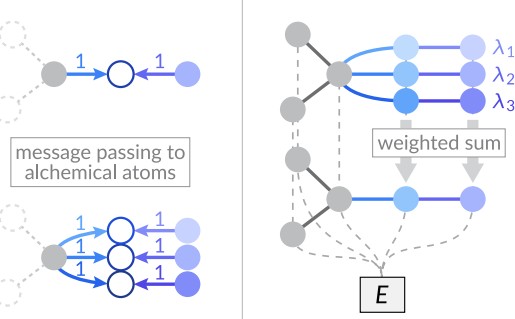

**Fig. 1 | Alchemical modification scheme for machine learning interatomic potentials. a** Alchemical graph augmentation: The relevant original atoms are split into alchemical atoms with different elemental identities, which are associated with alchemical weights $\lambda_i$. **b** Alchemical message passing: At the message aggregation step (Eq. (6)), each message contribution from neighboring atoms is weighted according to the asymmetric weighting scheme in Eq. (5). Only the weights from alchemical to non-alchemical atoms are weighted according to the alchemical weights of the source atoms to ensure consistency with the message-passing scheme in the original graph. **c** Alchemical energy readout: The energy contributions from the alchemical atoms are weighted according to their respective alchemical weights to yield the total energy prediction for the structure (denoted as $E$).

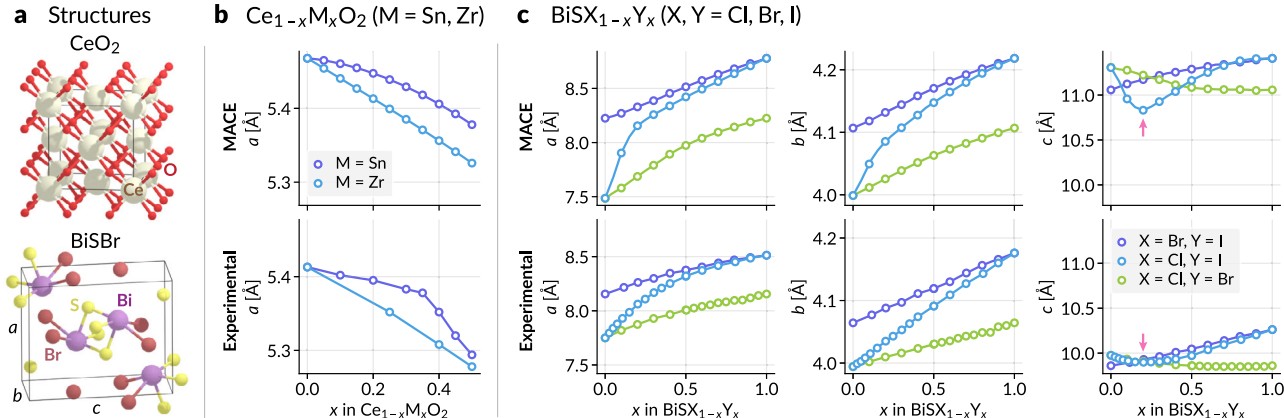

**Fig. 2 | Lattice parameters for solid solutions. a** The starting structures, $CeO_2$ and BiSBr, for solid solutions. **b** Lattice parameter $a$ for $Ce_{1-x}M_xO_2$ (M = Zr, Sn) as a function of the compositional weight $x$. **c** Lattice parameters $a$, $b$, and $c$ for $BiSX_{1-x}Y_x$ (X, Y = Cl, Br, I) as a function of $x$. The upper panels are the result of the

alchemically modified MACE-MP-0 medium model[15], and the lower panels are the experimental results from[48] and[49] for (**b**) and (**c**), respectively. Arrows in the rightmost panels indicate the composition with the minimum value of $c$. Source data are provided as a Source Data file.

This is a minimal prototype of MLIPs, and the state-of-the-art models use various additional mechanisms to enhance the expressivity to improve the fit to the training data. Although the alchemical modifications introduced in this work are based on this prototype, it can easily be integrated with such additional mechanisms, as further detailed in Section "Methods".

**Alchemical modification.** We now introduce the modifications to the input graph and the architecture of the MLIP model to allow the modeling of compositionally mixed structures with partial occupancies of atoms. The main idea is to augment the original graph with alchemical parts, creating an extra group of atoms or nodes for each compositional state to be modeled, and to modify the message passing scheme to keep it consistent with the baseline MLIP. First, we define the alchemical weights $\lambda = \{\lambda_\alpha\}_{\alpha=1}^k$ to assign the weights to each compositional state. For example, if we are modeling the mixed structure of LiCl, NaCl, and KCl with 20%, 30%, and 50% weights, respectively, the weights would be $\lambda = [0.2, 0.3, 0.5]$.

Now, we define an alchemical graph $\tilde{\mathcal{G}} = (\tilde{\mathcal{V}}, \tilde{\mathcal{E}})$ as an extension of an original graph $\mathcal{G} = (\mathcal{V}, \mathcal{E})$. For the previous example, we assume that we have an original graph representing the NaCl crystal structure. The construction is independent of the original elemental identities of the alchemical atoms, and only the atomic positions will be inherited. Each node in an alchemical graph is identified by a pair of indices, the original atom index $i$ and the alchemical index $\alpha$, and denoted by $(i, \alpha) \in \tilde{\mathcal{V}}$. All non-alchemical atoms (e.g., Cl), for which the element remains the same for all compositional states, are assigned with $\alpha = 0$ and the corresponding weight $\lambda_0 = 1$. Alchemical atoms are split into multiple nodes according to their compositional states (Fig. 1a). For example, the Na atom $i$ in the original graph is split into three nodes $(i, 1)$, $(i, 2)$, and $(i, 3)$, with elements $(Z_{(i,1)}, Z_{(i,2)}, Z_{(i,3)}) = $ (Li, Na, K). As such, the node features for alchemical atoms will be initialized with respective elemental embeddings. Then, we assign an alchemical weight $\lambda_\alpha$ to node $(i, \alpha)$. All other features, such as the positions of the atoms, are inherited from the original graph, e.g., $r_{(i,\alpha)} = r_i$.

Edges are connected between the alchemical graph nodes as in the original graph when either any the two endpoint nodes is non-alchemical (with weight index 0), or both nodes are in the same alchemical state (have the same weight index), i.e.,

$$\tilde{\mathcal{E}} = \{((i, \alpha), (j, \beta)) \mid (i, \alpha), (j, \beta) \in \tilde{\mathcal{V}} \wedge (i, j) \in \mathcal{E} \\ \wedge (\alpha = 0 \vee \beta = 0 \vee \alpha = \beta)\}. \quad (4)$$

This is in line with the dual topology paradigm widely utilized in the alchemical free energy literature[41–43], in which the atoms in the different alchemical states geometrically coexist but do not interact directly with each other. To model the scaled interaction between atoms in the alchemical graph, we introduce edge weights to scale the message contributions. Aldeghi and Coley[44] have proposed a similar idea in which they model the different topological assemblies of polymers by weighted (stochastic) edges between linkage atoms in monomers. Here, we use an asymmetrical weighting scheme given as

$$\omega_{\alpha\beta} = \begin{cases} \lambda_\beta & \text{if } \alpha = 0 \wedge \beta \neq 0 \\ 1 & \text{otherwise}, \end{cases} \quad (5)$$

i.e., only the message contributions from alchemical atoms to non-alchemical atoms are weighted by the alchemical weight of the source atom. This choice is based on the observation depicted in Fig. 1b. Since we are extending the original MLIP for alchemical compositions without modifying the learnable functions, we should ensure that the message passing is consistent with original graphs where all edge weights are implicitly 1. According to the expansion of alchemical atoms and the edge connection scheme, only the message passing from an alchemical atom to a non-alchemical atom is split into multiple pathways with respective alchemical node weights. Therefore, we utilize the alchemical node weights as the edge weights in this case, and the message aggregation scheme is modified from Eq. (1) as the weighted sum of the message contributions:

$$m_{(i,\alpha)}^{(t)} = \sum_{(j,\beta) \in \mathcal{N}((i,\alpha))} \omega_{\alpha\beta} M_t\left(h_{(i,\alpha)}^{(t)}, h_{(j,\beta)}^{(t)}, e_{ij}\right). \quad (6)$$

Finally, the readout for energy prediction (Eq. (3)) is modified as a weighted pooling of alchemical node contributions (Fig. 1c):

$$E = \sum_{(i,\alpha) \in \tilde{\mathcal{V}}} \lambda_\alpha R\left(h_{(i,\alpha)}^{(T)}\right). \quad (7)$$

Note that the same $M_t$ and $R$ functions as in Eqs. (1) and (3) are used, i.e., no trainable weights are modified. This modification scheme ensures two essential consistencies with the original MLIP scheme. First, when all of the alchemical elements are the same ($Z_{(i,\alpha)} = Z_i$) for each original atom and the alchemical weights sum up to 1 ($\sum_{\alpha=1}^k \lambda_\alpha = 1$), the predicted potential energy is the same with the original graph. Second,

when only one of the alchemical weights is 1 ($\lambda_\alpha = 1$), and the others are zero, the predicted potential energy is also the same as in the original graph with an elemental composition corresponding to $Z_\alpha$. These two consistencies in the limiting cases ensure the correct interpolation between compositional states, and although the argument here is based on the prototypical MLIP, the consistencies still hold when adapted to other architectures, as detailed in Section "Architecture-specific modifications" and Supplementary Information. We additionally explore alternative interpolation methods, including embedding interpolation, and compare their ability to interpolate the MLIP energy output in Supplementary Information.

### Representation of solid solution

**Lattice parameters.** First, we investigate whether our representation of a mixture of compositional states can be used to model solid solutions and to optimize their properties with respect to composition. Although many crystal properties can be tuned by the design choice of solid solutions[45], here we will use lattice parameters to probe the modeling ability. Empirically, the lattice parameters of solid solutions can be approximated by linear interpolation of those of constituent pure crystals with the corresponding compositional weights, as stated by Vegard's law[46,47]. Nevertheless, there are systems that exhibit significant positive or negative deviation from this idealized linear behavior, and we assess whether the proposed method is able to predict such trend.

First, the cell parameter for cubic $Ce_{1-x}M_xO_2$ solid solution exhibits linear behavior to $x$ when M = Zr, but shows a positive deviation with a kink for M = Sn[48]. We modeled this solid solution starting from the $CeO_2$ structure (Fig. 2a), splitting the Ce atoms into two alchemical states, Ce with weight $1-x$ and Zr or Sn with weight $x$, and optimizing the zero-temperature cell parameters by relaxing the unit cell. The alchemical scheme adapted for the universal MACE-MP-0 model[15] gives the correct linear behavior for M = Zr, and successfully identifies the positive deviation for M = Sn (Fig. 2b) although it fails to predict the kink. Further, we also model orthorhombic $BiSX_{1-x}Y_x$ (X, Y = Cl, Br, I) solid solutions, for which the lattice parameters $a$ (positive) and $c$ (negative with a local minimum) exhibit deviations from linearity[49]. We start from BiSBr structure (Fig. 2a) and split the Br atoms into two alchemical atoms of X and Y. The cell parameters are optimized with respective alchemical weights. For example, the $BiSCl_{1-x}I_x$ structure will have alchemical atoms Cl and I with alchemical weights of $\lambda = (1-x, x)$. The alchemical scheme with the MACE model correctly identifies the positive and negative deviations for $a$ and $c$, respectively, for X = Cl and Y = I (Fig. 2c). In particular, while the parameter $c$ is much larger than the experimental values (due to the inherent error in the original MLIP, itself likely arising from the underbinding nature of the PBE functional used to create the training data), the composition for the local minimum ($x \approx 0.2$) is accurately predicted. Although there is no direct correspondence between the alchemical weights and the stoichiometry of the solid solution, these results indicate that the representation developed here offers greater predictive accuracy compared to the naive estimate from Vegard's law. It is important to note that the current method assumes infinite disorder and thus neglects the effect of ionic ordering. In addition, because all the alchemical atoms are co-located in the position of the parent atom, the potential discrepancies among the fractional coordinates of substituent alchemical atoms are not taken into account.

**Compositional optimization.** Most MLIPs are designed to be end-to-end differentiable in order to obtain atomic forces and stress as gradients of the potential energy with respect to the positions $r_i$ and the strain tensor $\epsilon$, i.e., $F_i = -\partial E/\partial r_i$ and $\sigma = V^{-1}\partial E/\partial \epsilon$ where $V$ is the volume of the system. Gradient calculations are performed efficiently through the backward pass generated by automatic differentiation[50]. With our additional continuous representation of compositional states, the

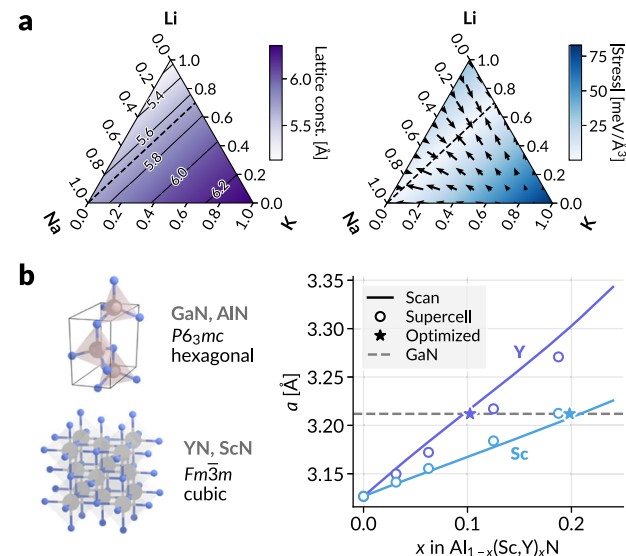

**Fig. 3 | Compositional optimization. a** Lattice parameter optimization for solid solutions of LiCl, NaCl, and KCl. The left panel shows optimized lattice parameters as a color gradient, obtained by relaxing the cell geometry for each compositional weight. The right panel displays hydrostatic stress, with color intensity representing stress magnitude and arrows indicating gradient direction, calculated by fixing the cell dimensions to those of NaCl. Since the energy output is end-to-end differentiable with respect to the alchemical weights, the composition can be optimized to match target cell dimensions (minimizing stress) by following these gradients. The compositions with cell parameters matching NaCl (left) and those obtained by minimizing stress (right), indicated by the dotted lines, align in both figures. **b** The optimization for the lattice-matching condition for solid solutions $Al_{1-x}Sc_xN$ and $Al_{1-x}Y_xN$ with GaN. The most stable polymorph structures are shown on the left. The plot on the right shows the cell dimension $a$ obtained by optimizing for each compositional weight (Scan), calculated from the corresponding supercell (Supercell), and the compositional weights optimized by gradient descent to match the $a$ value for GaN (Optimized). All results are obtained using the alchemically modified MACE-MP-0 medium model[15]. Source data are provided as a Source Data file.

alchemical weights $\boldsymbol{\lambda}$, we can also compute the gradients of the energy with respect to the composition $\partial E/\partial \boldsymbol{\lambda}$. Since the potential energy is defined up to constant, physically meaningful optimization targets are, in general, given by the energy difference or the gradient of the energy with respect to some system variables.

First, we consider a simple model: a solid solution of three alkali metal chlorides, LiCl, NaCl, and KCl. We fix the fractional coordinates of each atom and consider the cubic lattice constant as a function of alchemical (or compositional) weights of Li, Na, and K. To find a composition that matches a target lattice constant, we can enumerate a grid of compositions and relax the cell dimensions at each fixed composition to probe lattice constants over the compositional space (Fig. 3a, left). However, instead of this direct method, we can consider that the stress is minimized for the optimized structure and composition. Since our scheme is end-to-end differentiable, we can calculate the gradient of absolute hydrostatic stress $|\mathrm{tr}\,\boldsymbol{\sigma}|/3$ with respect to the composition where the lattice constant is fixed to the target value (Fig. 3a, right). Then, the optimal composition could be found by performing a gradient descent on the compositional space, offering a different approach to the design problem. This is more efficient because only a single gradient-based compositional optimization is required. In this case, since the size of Na is between Li and K, multiple optimal compositions exist on the compositional space.

Now, we apply this to a more realistic example, where we want to find the lattice-matching composition for solid solutions $Al_{1-x}Sc_xN$ and $Al_{1-x}Y_xN$ with GaN. The lattice-matched composition would facilitate the epitaxial growth of such solid solutions on the GaN substrate[51]. The

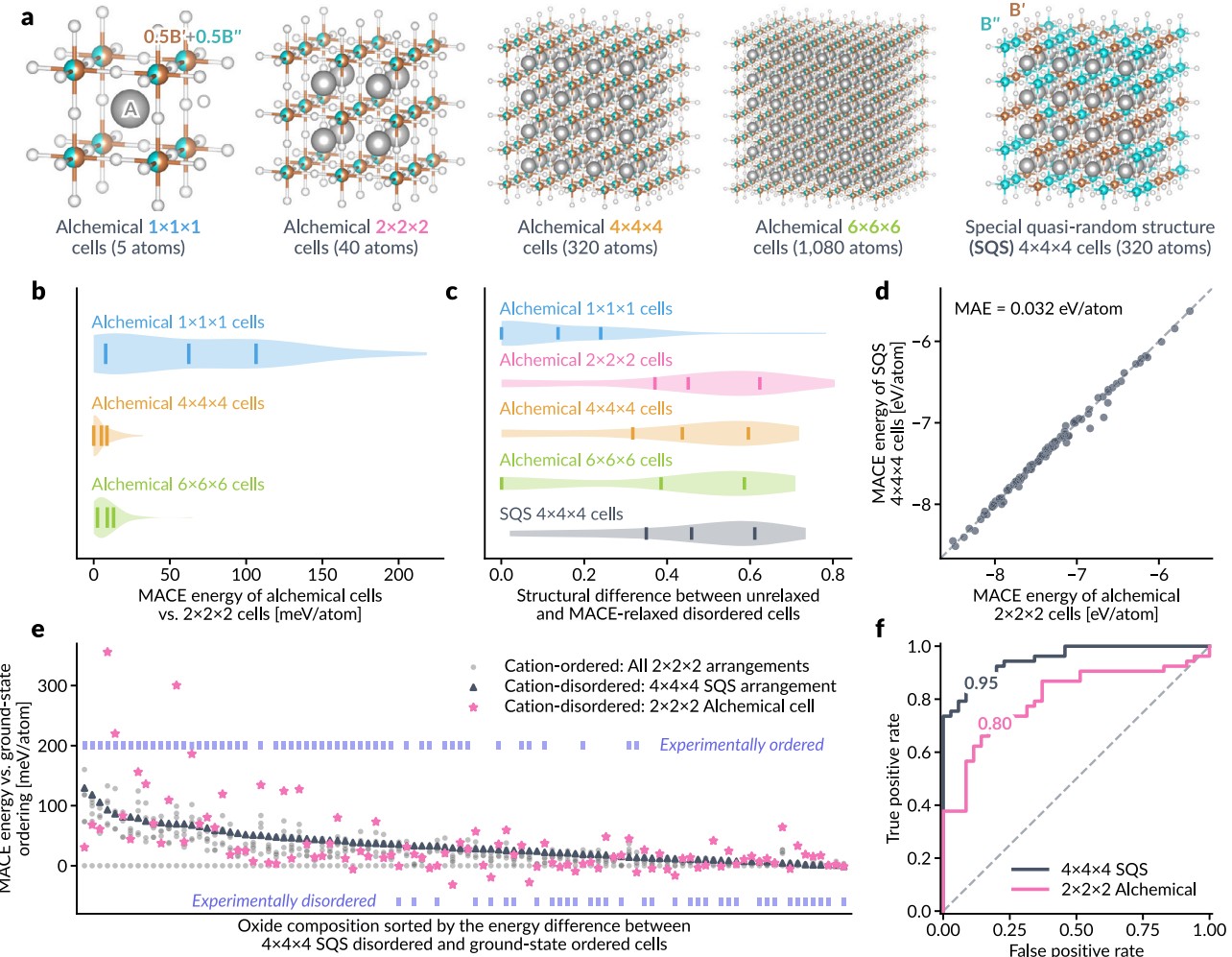

**Fig. 4 | Disordered energetics in multicomponent perovskite oxides. a** Crystal structure schematics for fully cation-disordered $A_2B'B''O_6$ perovskite oxide solid solutions, illustrating different alchemical supercell sizes and number of atoms, and representative 320-atom $4 \times 4 \times 4$ special quasirandom structures (SQSs). **b** MACE-relaxed energy differences between cation-disordered $2 \times 2 \times 2$ alchemical cells and smaller or larger supercells, evaluated across 100 $A_2B'B''O_6$ compositions from[52]. **c** Difference between the unrelaxed and MACE-relaxed structures for various alchemical cell sizes, including $4 \times 4 \times 4$ SQSs, quantified by cosine distance between local structure fingerprints. Vertical bars represent the first quartile, mean, and third quartile, respectively. **d** Comparison of MACE-relaxed energies for $2 \times 2 \times 2$ alchemical cells versus $4 \times 4 \times 4$ SQSs, with a mean absolute error (MAE) of 0.032 eV/atom. **e** MACE-relaxed energy differences among $2 \times 2 \times 2$ alchemical cells, $4 \times 4 \times 4$ SQSs, and all cation-ordered configurations with four B′ and four B″ cations on eight B sites in the $2 \times 2 \times 2$ supercell. Compositions are sorted by energy differences between $4 \times 4 \times 4$ SQS and lowest-energy ordered arrangements. Experimentally characterized ordered and disordered compositions[53] are marked in the upper and lower regions, respectively. **f** Receiver operating characteristic (ROC) curves for experimental order/disorder classification based on relative energy values of $4 \times 4 \times 4$ SQS or $2 \times 2 \times 2$ alchemical cells in (**e**) with respect to the lowest-energy cation-ordered arrangements, with area under the curve (AUC) values shown. All results are derived using the alchemically modified MACE-MP-0 medium model[15]. Source data are provided as a Source Data file.

objective is to determine a composition $x$ for each solid solution such that the cell parameter $a$ of the lattice matches the value for the GaN structure. Although GaN and AlN possess a hexagonal lattice (space group $P6_3mc$), pure YN and ScN have a cubic lattice (space group $Fm\bar{3}m$), which means that one cannot simply interpolate between the cell parameters of the constituent compounds to infer those of solid solutions. Here, we fix the cell parameter $a$ for the hexagonal lattice, and we optimize the relevant stress components with respect to the cell parameter $c$ as well as the Al/Sc or Al/Y composition (see Section "Representation of solid solution") because the doped AlN would result in different $c/a$ ratio. Results in Fig. 3b show that the optimized compositions are $x \approx 0.1$ (Y) and $x \approx 0.2$ (Sc), and are in good agreement with the forward scan result, where the relaxed cell parameters are measured while scanning for various $x$ values. Furthermore, we created a $4 \times 4 \times 4$ supercell of AlN and randomly switched some Al atoms to Sc or Y atoms to match the target composition and measured the unit cell parameters. These results match well with the scan results over alchemical unit cell

compositions, which indicates that the methodology in the current work can also be regarded as a computationally efficient compact representation of the supercell with compositional disorder.

**Disorder energetics.** The high computational efficiency and accuracy of alchemically modified MLIPs for modeling disordered solid solutions are further validated by examining a dataset of $A_2B'B''O_6$ multicomponent perovskites in our recent high-throughput studies[52,53]. Notably, the thermodynamic preference of an $A_2B'B''O_6$ perovskite to adopt either cation-ordered or cation-disordered structures depends on the difference between formation energetics of various cation-ordered configurations and those of cation-disordered solid solutions[52]. For ordered structures, the formation energetics across all possible symmetrically inequivalent cation arrangements can serve as physics-informed descriptors to predict the thermodynamic tendency towards experimental cation disorder[53]. While DFT is computationally prohibitive for evaluating formation energetics of various enumerated cation-ordered

atomic arrangements, we have shown that symmetry-aware equivariant graph neural networks, including equivariant MLIPs, provide efficient and accurate surrogates for assessing ordering-dependent thermodynamic stability in multicomponent perovskite oxides[52].

Here, we extend our previous analysis to directly examine the formation energetics of fully cation-disordered $A_2B'B''O_6$ solid solutions with partial B site occupancies of 0.5 B′ and 0.5 B″. Traditionally, special quasirandom structures (SQS)[54,55], which optimize elemental placements within a supercell to mimic the cluster vectors of random alloys, have been widely used to study disordered solid solutions. While the SQS approach provides a systematic approach to model disordered structures, it requires large supercells to avoid correlations across periodic boundaries and relies on optimization routines such as Monte Carlo simulations[56], limiting its feasibility for high-throughput studies. Given the efficiency of alchemically modified MLIPs in representing disorder through partial elemental occupancies, we compare alchemical unit cell modeling of perovskite solid solutions to SQS cells using baseline MLIPs for disorder modeling.

Starting from the base ordered perovskite $ABO_3$ structure, we split the B atom into two alchemical species, B′ and B″, each assigned an alchemical weight of 0.5. We then generate $N \times N \times N$ ($N = 1, 2, 4, 6$) alchemical supercells and $4 \times 4 \times 4$ SQS supercells (Fig. 4a), optimizing each cell using alchemically modified MACE-MP-0 and baseline MACE-MP-0 models. For alchemical supercells, the relaxed cell energy per atom pleateaus at the $2 \times 2 \times 2$ supercell, while the unit cell ($1 \times 1 \times 1$) exhibits notably higher energy compared to larger supercells (Fig. 4b). The structural differences between unrelaxed and relaxed disordered cells, shown in Fig. 4c, quantified using cosine distances of local structural fingerprints[53,57], reveal that the alchemical unit cell relaxes only slightly, whereas larger alchemical supercells and SQS cells show more significant differences between their corresponding unrelaxed and MLIP-relaxed structures. As noted in previous works[52,53], crystallographic sites undergo substantial distortion during relaxation, such as octahedral tilting and Jahn−Teller distortions[58], which are typically beyond the periodicity of a perovskite unit cell and thus can hardly be captured by modeling a single unit cell. Given that the $2 \times 2 \times 2$ supercell yields results similar to those of larger alchemical supercells, we proceed with further analysis using the 40-atom $2 \times 2 \times 2$ alchemical supercell.

As shown in Fig. 4d, the optimized single-point energies from the alchemical $2 \times 2 \times 2$ supercell align well with those from the $4 \times 4 \times 4$ SQS supercell, with a mean absolute error (MAE) of 0.032 eV/atom. Since the preference for cation-ordered and cation-disordered configurations depends on the relative formation energetics of each, we further compare energy values with all symmetrically inequivalent cation-ordered configurations in the $2 \times 2 \times 2$ supercell, obtained by enumerating four B′ and four B″ cations occupying eight B sites[53]. The results in Fig. 4e show the relative energies of all considered structures, aligned with the ground-state (lowest-energy) cation-ordered structure energy as the reference. As previously discussed in refs. 52,53, we observe that experimentally observed ordered compositions exhibit significant difference between the ground-state ordered configuration energy and other configurations, whereas experimentally cation-disordered compositions show similar energies among different configurations. The relative energy of the disordered SQS supercell provides a useful metric for characterizing experimental order/disorder, as seen by the separation of ordered and disordered compositions when sorting the oxide compositions by the SQS energy. Although the $2 \times 2 \times 2$ alchemical supercell energies show more stochasticity, they follow the same trend as the relative energies of the SQS. This is further supported by the receiver operating characteristic (ROC) curves for experimental order/disorder classification based on relative energy values (Fig. 4f), where $4 \times 4 \times 4$ SQS cell energies provide excellent classification with an area under the curve (AUC) of 0.95, while the alchemical $2 \times 2 \times 2$ cell energies achieve reasonably good experimental order/disorder classification with an AUC of 0.80. The likely

source of this difference is that for ions of very different sizes, local structural distortions are related to local chemical ordering, but the use of an average structure imposed by the alchemical method fails to produce local distortions that SQS captures well.

Hence, based on these results, we conclude that the alchemical modification of MLIPs offers a scalable approach for disorder modeling, as demonstrated with this multicomponent perovskite oxide dataset. The alchemical $2 \times 2 \times 2$ supercells provide reasonable accuracy for experimental disorder classification, while using only 1/8 of the atoms in the $4 \times 4 \times 4$ SQS supercells. Unlike SQS, these alchemical supercells can be obtained without the need for additional annealing steps for configuration generation. The results were achieved by modifying off-the-shelf pre-trained MLIPs and could be further fine-tuned to improve energy prediction and order-disorder classification. They may also be adapted for other material systems, including compositionally complex alloys and ceramics.

We also note that our approach shares similarities with the Virtual Crystal Approximation (VCA)[59–61], a traditional approach in modeling solid solutions with partial elemental site occupancy. VCA relies on two assumptions: (1) geometry: the solid solution is represented by an averaged structure where crystallographic sites are randomly occupied by different elements, disregarding local ordering; and (2) interaction: the random occupancy is approximated by compositionally weighted average of atomic pseudopotentials. Our method adopts the first assumption, making it subject to the same geometric limitations, such as the elements should be of similar size, occupy comparable positions, and local disorder effects should be minimal. However, the practical limitations of VCA mainly arise from what could be described as pseudopotential alchemy, where accuracy depends heavily on carefully tuning pseudopotential parameters like radial cutoffs and electronic configurations (core/valence). In contrast, our method sidesteps these challenges: MLIPs replace electronic structure calculations with iterative message-passing between node and edge features. Built-in regularization from training scheme and model architecture help ensure that results remain within a physically reasonable range, reducing the need for extensive manual parameter adjustments.

## Free energy calculations

**Free energy calculations.** Here, we utilize the nonequilibrium switching method, where the Hamiltonian depends on a progression parameter $\lambda \in [0, 1]$ so that it interpolates between the initial Hamiltonian $H_i = H(\lambda = 0)$ and the final Hamiltonian $H_f = H(\lambda = 1)$. Assuming the NVT ensemble, the reversible work is given via the TI equation[62]:

$$\Delta F = W_{i \to f}^{rev} = \int_0^1 d\lambda \left\langle \frac{\partial H}{\partial \lambda} \right\rangle_\lambda. \tag{8}$$

We now consider a finite-time process in which $\lambda$ is switched from 0 at time $t_i$ to 1 at time $t_f$. The irreversible work done by switching the Hamiltonian is

$$W_{i \to f}^{irrev} = \int_{t_i}^{t_f} dt \frac{d\lambda}{dt} \frac{\partial H}{\partial \lambda} = W_{i \to f}^{rev} + E_{i \to f}^{diss}, \tag{9}$$

where $E_{i \to f}^{diss}$ is the dissipated energy. In a linear-response regime, it can be shown[63,64] that the dissipated energy for the forward and backward path is the same when averaged over the transition path ensemble, i.e.,

$$\overline{E^{diss}} = \overline{E_{i \to f}^{diss}} = \overline{E_{f \to i}^{diss}} = \frac{1}{2} \left( \overline{W_{i \to f}^{irrev}} + \overline{W_{f \to i}^{irrev}} \right). \tag{10}$$

Then, the free energy difference can be computed as

$$\Delta F = \frac{1}{2} \left( \overline{W_{i \to f}^{irrev}} - \overline{W_{f \to i}^{irrev}} \right). \tag{11}$$

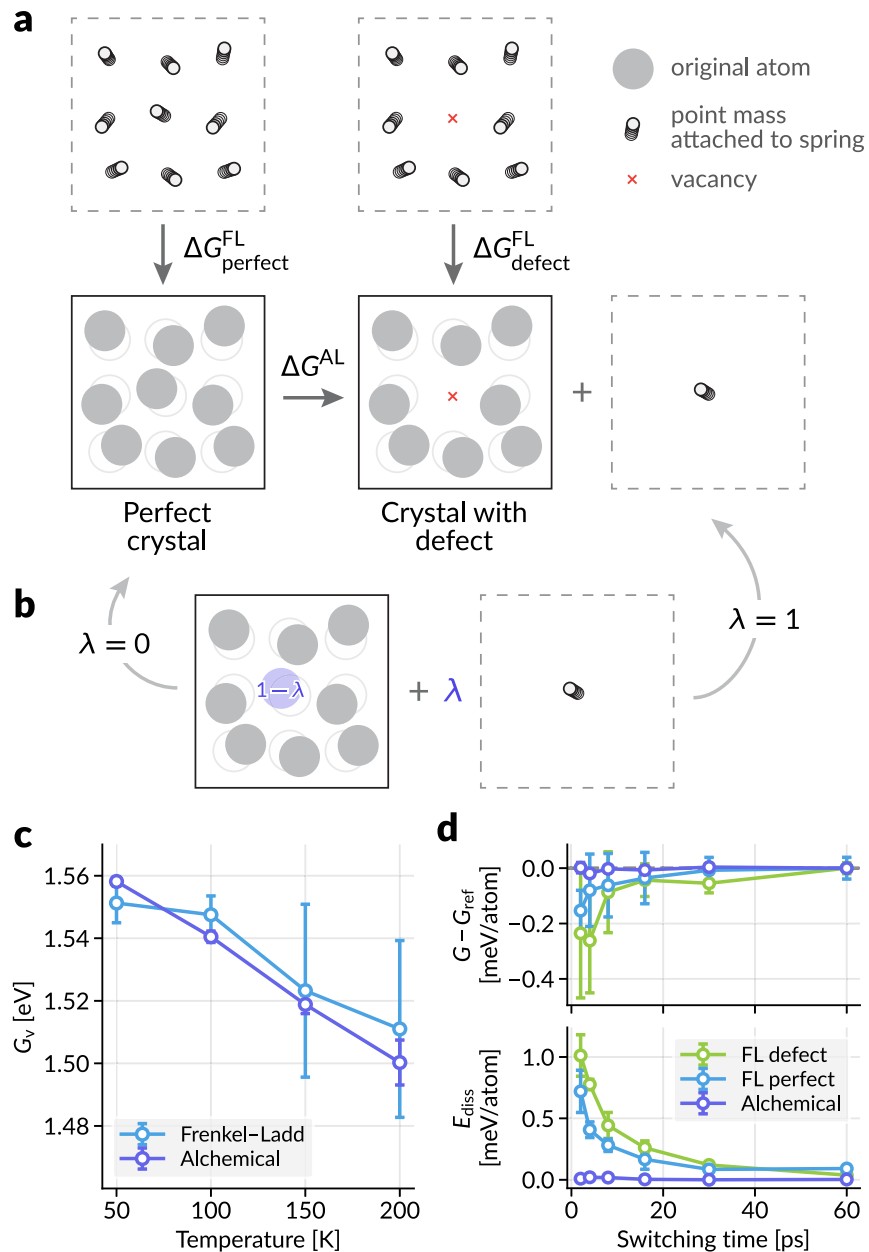

**Fig. 5 | Free energy of vacancy formation in BCC iron. a** Transformations used to determine the Gibbs free energy of the perfect crystal and the crystal with a defect. The alchemical pathway used here transforms the perfect crystal into the crystal with a defect and a single atom attached to a spring to avoid diffusion. $\Delta G^{\mathrm{FL}}$ and $\Delta G^{\mathrm{AL}}$ represent the Gibbs free energy changes calculated via the Frenkel–Ladd and alchemical pathways, respectively. **b** The intermediate state parameterized by $\lambda$ for the alchemical pathway in (**a**). The atom to be removed is assigned an alchemical weight of $1 - \lambda$, and the energy of the harmonic oscillator is scaled by $\lambda$. **c** The free energy of vacancy (Eq. (12)) computed by the Frenkel–Ladd path and alchemical path. **d** Statistical efficiency for the Frenkel–Ladd paths and alchemical path at 100 K against the switching time. Upper panel shows the deviation of Gibbs free energy from the reference value at the longest switching time (60 ps), and the lower panel shows average dissipated energies (Eq. (10)). For panels **c** and **d**, each data point represents the average of four statistically independent simulations, with standard deviations shown as error bars. Source data are provided as a Source Data file.

Often, the Hamiltonian is parametrized by the linear interpolation of the two endpoints, i.e., $H(\lambda) = (1 - \lambda)H_i + \lambda H_f$, to simplify the calculation of the gradient term $\partial H / \partial \lambda$ in Eqs. (8) and (9): $\partial H / \partial \lambda = H_f - H_i$. However, we note that in our case, the system Hamiltonian can be parametrized by the alchemical weights, and $\partial H / \partial \lambda$ can be calculated straightforwardly using automatic differentiation[50] on the MLIP. This method proves to be more efficient than linear interpolation as it obviates the need to repeat calculations for non-changing atoms. We compare computational efficiencies and the resultant free energy calculations in Supplementary Information.

**Free energy of vacancy formation.** Accurate evaluation of the free energy of a point defect is important for characterizing its thermodynamic stability[65]. Here, we calculate the Gibbs free energy of vacancy defined as

$$G_v = G_{\mathrm{defect}} - \frac{N-1}{N} G_{\mathrm{perfect}}, \qquad (12)$$

where $G_{\mathrm{defect}}$ and $G_{\mathrm{perfect}}$ are the Gibbs free energies of crystal with and without a point defect, and $N$ is the number of atoms in the perfect crystal. Because the vacancy diffuses at high temperatures, it is

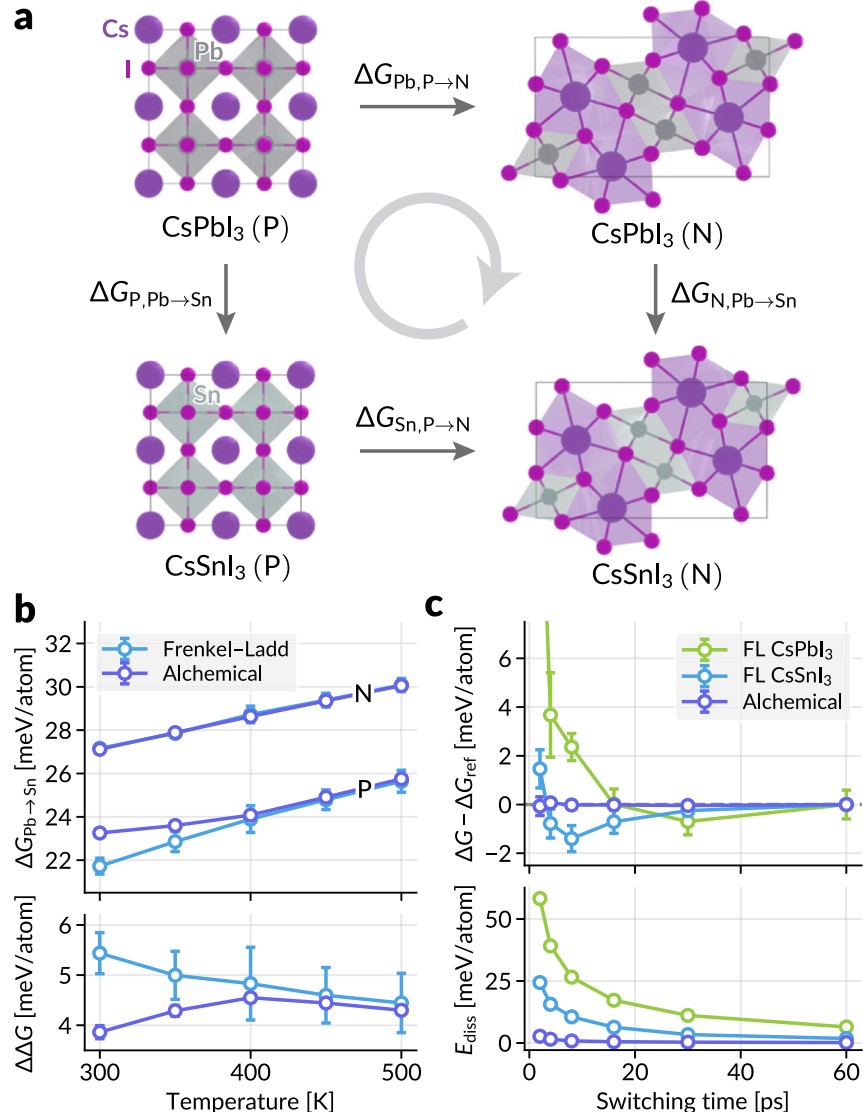

**Fig. 6 | Alchemical free energy simulations. a** The thermodynamic cycle considered in this work, consisting of perovskite (P) and non-perovskite (N) phases of $CsPbI_3$ and $CsSnI_3$. $\Delta G$ values are labeled by phase (P or N) and composition; for instance, $\Delta G_{N,Pb\to Sn}$ indicates the gibbs free energy change for the non-perovskite phase from $CsPbI_3$ to $CsSnI_3$. **b** Upper panel: the alchemical free energy of Pb→Sn conversion in both phases, plotted against the simulation temperature. Lower panel: the $\Delta\Delta G$ values in Eq. (13) computed from the results in the upper panel. The deviations between the two methods at lower temperatures result from the phase transformation between the perovskite phases. **c** Statistical efficiency for the Frenkel–Ladd paths and alchemical path of $CsPbI_3$ to $CsSnI_3$ transformation for P phase, plotted against the switching time. Upper panel shows the deviation of Gibbs free energy from the reference value at the longest switching time (60 ps), and the lower panel shows average dissipated energies (Eq. (10)). For panels **b** and **c**, each data point represents the average of four statistically independent simulations, with standard deviations shown as error bars. Source data are provided as a Source Data file.

common to first evaluate the Gibbs free energies at low temperatures in which the vacancy is fixed at one site[66] and extend the calculation by considering the temperature dependence of Gibbs free energy[67]. Hence, we will focus on determining the Gibbs free energy of vacancy in BCC iron at low temperatures and compare the result with Gibbs free energies determined using the Frenkel–Ladd path[68], which is commonly used in nonequilibrium calculations[33,64]. In the Frenkel–Ladd path, the crystal structure is switched from and to a system of independent harmonic oscillators with the same equilibrium positions (the Einstein crystal), for which we can calculate the exact free energy. See Section "Free energy calculations" for more details on the reference calculation.

We introduce a new alchemical path for determining the free energy of vacancy, as depicted in Fig. 5a. While the previous examples of our method were restricted to cases where $\sum_\alpha \lambda_\alpha = 1$, we can lift this

restriction to create or annihilate atoms in a system alchemically. In this case, we assign alchemical weight $\lambda_1 = 1 - \lambda$ to the atom in the vacancy site and switch the weight from 1 to 0 ($\lambda$ from 0 to 1) over the simulation to make it continuously disappear from the system. At the same time, we add the harmonic oscillator term to the atom position with weight $\lambda$, so that through the alchemical conversion from $\lambda = 0$ to 1 transforms the perfect crystal into a crystal with defect and a harmonic oscillator (Fig. 5b). Through nonequilibrium switching simulations, we can obtain the alchemical free energy difference $\Delta G^{AL}$ (Eq. (23)). We now compare the free energy of vacancy (Eq. (12)) obtained from both Frenkel–Ladd calculations ($G_{defect}^{FL}$ and $G_{perfect}^{FL}$) and with alchemical free energy calculations ($\Delta G^{AL}$ and $G_{perfect}^{FL}$).

The results in Fig. 5c show that $G_v$ calculated by the proposed alchemical free energy method is comparable to that from the reference Frenkel–Ladd calculations, while offering more consistent results

with much smaller standard deviations when using the same switching time steps. We further investigate the statistical efficiency of the switching paths at 100 K by evaluating the convergence of $\Delta G$, taking the longest switching time result as its reference, as well as the dissipated energy $E_{diss}$ (Eq. (10)) in Fig. 5d. The alchemical pathway offers much faster convergence, with minimal average energy deviations ( < 0.02 meV/atom) from the reference value, even at a very short switching time of 2 ps (1000 MD steps).

**Alchemical free energy calculations.** Now, we examine the effectiveness of the proposed alchemical scheme in the calculation of alchemical free energy difference associated with the change in the elemental identities of the atoms. We use halide perovskites $CsPbI_3$ and $CsSnI_3$ as our model system, which have been studied using MLIPs (e.g.,[69]) and classical force fields (e.g.,[70,71]). Both $CsPbI_3$ and $CsSnI_3$ exhibit three photoactive perovskite phases, $\alpha$ (cubic, $Pm\bar{3}m$), $\beta$ (tetragonal, $P4/mbm$), and $\gamma$ (orthorhombic, $Pnma$), in decreasing order of temperature window of stability. However, they also possess a photoinactive non-perovskite polymorph, $\delta$ (orthorhombic, $Pnma$), which is the most stable phase at room temperature[72]. Here, we analyze the difference in the relative stabilities of perovskite (P) and non-perovskite (N) phases as shown in the thermodynamic cycle in Fig. 6a. The direct computation of the free energy of phase transformation, $\Delta G_{Pb,P\to N}$ and $\Delta G_{Sn,P\to N}$, may require enhanced sampling simulations with tailored collective variables or nonequilibrium simulations (the Frenkel–Ladd paths) with longer simulation time until convergence. The alchemical path enables the calculation of $\Delta G_{P,Pb\to Sn}$ and $\Delta G_{N,Pb\to Sn}$. Since the two types of free energy differences are linked by

$$\begin{aligned}\Delta\Delta G &= \Delta G_{N,\,Pb\to Sn} - \Delta G_{P,\,Pb\to Sn}\\ &= \Delta G_{Sn,\,P\to N} - \Delta G_{Pb,\,P\to N},\end{aligned} \tag{13}$$

we can compute the difference in the relative stability of phases upon compositional changes, or we can calculate either of the free energies of phase transformation if another is already known.

For the alchemical free energy simulation, starting from the $CsPbI_3$ structure, the Cs and I atoms remain as non-alchemical atoms, and the Pb atoms are divided into alchemical atoms, Pb and Sn, with alchemical weights $\lambda_1 = 1 - \lambda$ and $\lambda_2 = \lambda$. Then, switching $\lambda$ from 0 to 1 continuously transforms the $CsPbI_3$ structure into the $CsSnI_3$ structure. Refer to Section "Free energy calculations" for more details on the alchemical free energy calculation settings and result analysis required to obtain the Gibbs free energies.

First, we compare the Gibbs free energy of compositional change from two methods: $\Delta G_{P/N,Pb\to Sn}^{AL}$ from the alchemical path and $\Delta G_{P/N,Pb\to Sn}^{FL} = G_{P/N,Sn}^{FL} - G_{P/N,Pb}^{FL}$ from the Frenkel–Ladd path for each composition. The results in Fig. 6b indicate that the two calculation results coincide well except for the slight deviation in the perovskite phase for temperatures lower than 400 K. The deviation may occur from the phase transformation between perovskite phases of $CsPbI_3$ (i.e., $\alpha \to \beta$). The Frenkel–Ladd path is simulated in the fixed cell (NVT) of the respective $\alpha$ phase, whereas the alchemical path is simulated in the NPT ensemble, in which phase transformations can occur. Given that the $\beta$ phase is more stable than the $\alpha$ phase for $CsPbI_3$ at low temperatures, $\Delta G_{Pb\to Sn}^{AL}$ is expected to be larger than $\Delta G_{Pb\to Sn}^{FL}$, as in Fig. 6b. See Supplementary Information for further discussion. The calculation of $\Delta\Delta G$ (Eq. (13)) also shows that the two results are well matched at higher temperatures, while the alchemical path provides smaller standard deviations from multiple runs.

Similarly to the previous example, we analyzed the convergence of the Gibbs free energy and the energy dissipation for the alchemical path for the perovskite phase at 400 K by changing the switching time for nonequilibrium simulations. Fig. 6c shows that, similar to the previous result, the alchemical path provides much faster convergence

than the Frenkel–Ladd path. This result confirms that the phase space overlap between the two same phase structures with different compositions is much more significant than that between the atomic structures and the Einstein crystals, which enables much more efficient free energy simulations.

## Discussion

The alchemical modification of MLIP introduced in this work allows a smooth interpolation between structures with two or more different compositions. Building upon a prototypical construction of MLIP, we modified the input graph, message passing scheme, and readout layers to alchemically weight the different compositional states. Although this modification can be generalized to various classes of MLIPs, it is particularly efficient when integrated with MACE because of its construction of many-body features from two-body messages (see Section "Architecture-specific modifications").

We first applied the scheme to the modeling of solid solutions. Although there is no theoretical relationship between the stoichiometry and the alchemical weights, the results showed that it could model the nonlinear deviations of cell parameters in some solid solutions. The end-to-end differentiability of the model with respect to the alchemical weights enabled the optimization of composition to match the desired cell parameters. The alchemical modification also provides a scalable, efficient method for characterizing order and disorder, as demonstrated in multicomponent perovskite oxides, achieving accuracy comparable to SQS with fewer number of atoms and no optimization needed for structure generation. Furthermore, the alchemical weights allow smooth creation or annihilation of atoms, or the change in atom types, enabling the calculation of free energy differences between two compositional states. We demonstrated that the free energy of vacancy in BCC iron and the relative phase stabilities of the perovskite and non-perovskite phases of $CsPbI_3$ and $CsSnI_3$ could be calculated much more efficiently than the widely utilized Frenkel–Ladd path. It is worth noting that, unlike the modeling of solid solutions, alchemical free energy calculations conducted here are theoretically exact when reaching convergence.

Overall, the proposed method enables efficient modeling of composition-related properties with sufficient consistency within the underlying MLIP. Beyond the aforementioned lack of theoretical ground on the connection between alchemical weights and stoichiometric coefficients and convergence questions that are universal to thermodynamic integration methods, inaccuracies emerge primarily from the MLIP. In particular, there are two sources of error: (1) the discrepancies between the MLIP and the DFT calculations and (2) the inaccuracy of the underlying DFT calculations. Since most universal MLIPs are trained on the energies and derivatives from the relaxation trajectory, the relative error around the energy minima would be small. This implies that the former error would also be small when performing free energy calculations for systems with a sufficient number of similar structures in the materials database. Fine-tuning the MLIP using the DFT data from relevant compositional space would alleviate the former error. One can also utilize free energy perturbation methods[73] to calculate the free energy from a more accurate Hamiltonian to reduce both types of errors. We also note that differentiable simulations[74,75] could be used to fine-tune the MLIP to match either the cell parameters resulting from the relaxation trajectory or the free energy differences from the MD simulations to their desired values, to mitigate both sources of errors.

While we devised the alchemical scheme with fixed elemental identities and $\lambda$ representing the occupancies of different alchemical atoms to align with our goal of leveraging pre-trained MLIPs, we note that interpolating the elemental identities of atoms, i.e., coupling $\lambda$ to atomic numbers, is also a promising direction that connects with the quantum alchemy literature. Although pre-trained embeddings may not

be ideal for this, they could be fine-tuned by alchemical force matching with $\partial E/\partial \lambda$ derived from quantum alchemy[22,76,77], possibly using analytical gradients[78–80], given that the baseline MLIP is end-to-end differentiable with respect to embeddings. Learning an MLIP-based representation consistent with quantum alchemy might offer a well-regularized approximation of the physical state for the TI calculation. While this alternative scheme could improve the physical relevance of alchemical degrees of freedom, it is incompatible with the current approach and remains a prospect for future work. Beyond the applications demonstrated in this work, we expect that the gradient of the physical observables with respect to the composition or elemental identities would hold particular importance to the generative modeling of molecules and materials systems. We envisage that further works, integrated with the discrete sampling literature[81,82], will utilize the alchemical degrees of freedom in MLIPs for such modeling applications.

## Methods

### Architecture-specific modifications

In MACE, the atomic basis $A_i^{(t)}$ is constructed by pooling the two-body features over the neighbors as in Eq. (14) (Eq. (8) in the original paper[10]). The modification to message passing in Eq. (6) is implemented by multiplying the edge weights $\omega_{\alpha\beta}$ (Eq. (5)) to the summands of the message aggregation as in Eq. (15):

$$A_{i,kl_3m_3}^{(t)} = \sum_{l_1m_1,l_2m_2} C_{l_1m_1,l_2m_2}^{l_3m_3} \sum_{j \in \mathcal{N}(i)} R_{kl_1l_2l_3}^{(t)}(r_{ji}) Y_{l_1}^{m_1}(\hat{r}_{ji}) \sum_{\bar{k}} W_{k\bar{k}l_2}^{(t)} h_{j,\bar{k}l_2m_2}^{(t)},$$
(14)

$$A_{(i,\alpha),kl_3m_3}^{(t)} = \sum_{l_1m_1,l_2m_2} C_{l_1m_1,l_2m_2}^{l_3m_3} \sum_{(j,\beta) \in \mathcal{N}((i,\alpha))} \omega_{\alpha\beta} R_{kl_1l_2l_3}^{(t)}(r_{ji}) Y_{l_1}^{m_1}(\hat{r}_{ji}) \sum_{\bar{k}} W_{k\bar{k}l_2}^{(t)} h_{(j,\beta),\bar{k}l_2m_2}^{(t)}.$$
(15)

The original readout mechanism is the sum of site energies over all the outputs of readout layers $\mathcal{R}_t(\cdot)$ (Eq. (16)). We implement the alchemical readout in Eq. (7) as the weighted sum of alchemical site energies as in Eq. (17):

$$E = \sum_{i \in \mathcal{V}} E_i = \sum_{i \in \mathcal{V}} \sum_{t=0}^{T} \mathcal{R}_t\left(h_i^{(t)}\right),$$
(16)

$$E = \sum_{(i,\alpha) \in \bar{\mathcal{V}}} \lambda_\alpha E_{(i,\alpha)} = \sum_{(i,\alpha) \in \bar{\mathcal{V}}} \lambda_\alpha \sum_{t=0}^{T} \mathcal{R}_t\left(h_{(i,\alpha)}^{(t)}\right).$$
(17)

### Representation of solid solution

We used the MACE-MP-0 medium model[15] for the experiments in Section "Representation of solid solution". Fast Inertial Relaxation Engine (FIRE) algorithm[83], as implemented in the Atomic Simulation Environment (ASE) 3.22.1 package[84], was used to conduct geometry relaxations under fixed composition.

**Compositional optimization.** For the optimization for lattice-matching composition for solid solutions $Al_{1-x}Sc_xN$ and $Al_{1-x}Y_xN$ with GaN, we used $|\sigma_{xx} + \sigma_{yy}|$ as the optimization target to find the matching condition for cell parameter $a$. We used gradient descent with learning rates 0.01 and 0.005 for $c$ and alchemical weights $\lambda$, respectively. We initialized $c$ with the value from the optimized GaN structure and $\lambda$ with [1, 0]. The gradient of $\lambda$ was projected onto the line $\lambda_1 + \lambda_2 = 1$ by subtracting the mean value at each optimization step.

In general, when alchemical weights $\lambda$ represent the compositional states, the weights should add up to 1 and the individual weights should be non-negative, i.e., the weights are element of the compositional simplex $\Delta^{k-1} = \{\lambda \in \mathbb{R}^k \mid \sum_{\alpha=1}^k \lambda_\alpha = 1, \lambda_\alpha \geq 0\}$. We can perform gradient-based constrained optimization for the minimization target $\mathcal{L}(\lambda)$ on the simplex by utilizing the exponentiated gradient descent method[85,86] with the update rule given as

$$\lambda_\alpha^{(t+1)} = \frac{\lambda_\alpha^{(t)} \exp(-\eta \cdot \partial\mathcal{L}/\partial\lambda_\alpha)}{\sum_\beta \lambda_\beta^{(t)} \exp(-\eta \cdot \partial\mathcal{L}/\partial\lambda_\beta)},$$
(18)

where $\eta$ is the learning rate.

**Disordered energetics.** We obtained $2 \times 2 \times 2$ cation-ordered arrangements and experimental order/disorder label from the dataset reported in[52,53] on ordering of multicomponent perovskite oxides $A_2B'B''O_6$. We used the icet 3.0 package[87] to optimize and generate $4 \times 4 \times 4$ cation-disordered SQSs. Structural differences are identified by pooling localized fingerprints, generated using `OPSiteFingerprint` from matminer 0.9.3[88], across all atoms to create crystal-level feature vectors[53,57]. The difference between two crystal features, $x$ and $y$, is quantified as the cosine distance: $d(x,y) = 1 - x^\top y/(\|x\|\|y\|) \in [0, 2]$.

### Free energy calculations

We used the MACE-MP-0 small model[15] for the experiments in Section "Representation of solid solution". MD integrations are performed with corresponding implementations in ASE[84], and a time step of 2 fs was used. The characteristic time scales of $\tau_T = 25$ fs and $\tau_P = 75$ fs were used for all the thermostats and barostats, respectively. Each simulation was initialized with energy minimization (with or without the cell fixed) using the FIRE algorithm[83] and sampling the momenta from the Maxwell–Boltzmann distribution at the given temperature. The center of mass of the system was fixed for all MD simulations. For all nonequilibrium simulations, we used the progress parameter scheduling of

$$\lambda(\tau) = \tau^5(70\tau^4 - 315\tau^3 + 540\tau^2 - 420\tau + 126),$$
(19)

where $\tau = t/t_{switch} \in [0, 1]$ is the normalized switching time progress. Instead of linear $\lambda(\tau) = \tau$, the scheme in Eq. (19) was used because the slope $d\lambda/dt$ vanishes at both ends and reduces the energy dissipation[89]. All free energy calculation results were averaged over four statistically independent simulations, and their standard deviations are reported as error bars in Figs. 5 and 6.

**Frenkel–Ladd path.** Here, we adapted the procedure arranged in Menon et al.[33]. First, the system was equilibrated for 60 ps under the NPT ensemble with the pressure of $P = 1$ atm by the Berendsen thermostat and homogeneous Berendsen barostat[90]. The average cell volume $\langle V \rangle$ was calculated during the last 40 ps of the simulation. The spring constants for the Einstein crystal were estimated from the mean-squared displacement (MSD) of the atoms under the fixed system volume as $k_i = 3k_BT/\langle(\Delta r_i)^2\rangle$[64]. The fixed volume system was simulated for 100 ps under the NVT ensemble by the Langevin thermostat with $\gamma = 1/\tau_T$[91], and the MSD values were computed over the last 60 ps of the simulation. The MSD values were averaged and reassigned to the symmetrically equivalent atoms to reduce the variance before determining the spring constants. Finally, a nonequilibrium simulation of the Frenkel–Ladd path with the determined spring constants was conducted under the NVT ensemble using the Langevin thermostat. The system was equilibrated at $\lambda = 0$ for 40 ps, switched from $\lambda = 0$ to 1 with the schedule in Eq. (19) for 60 ps, equilibrated again at $\lambda = 1$ for 40 ps, and switched back from $\lambda = 1$ to 0 for 60 ps.

The Helmholtz free energy for independent harmonic oscillators (Einstein crystal) with angular frequencies $\omega_i = (k_i/m_i)^{1/2}$ is given as

$$F_E(N, V, T) = 3k_B T \sum_{i=1}^{N} \ln\left(\frac{\hbar\omega_i}{k_B T}\right). \quad (20)$$

Since the center of mass of the system is fixed, the following finite-size correction associated with the Frenkel–Ladd path was applied:

$$\Delta F_{CM} = k_B T \left[\ln\frac{N_{WS}}{V} + \frac{3}{2}\ln\left(2\pi k_B T \sum_{i=1}^{N} \frac{\mu_i^2}{k_i}\right)\right], \quad (21)$$

where $\mu_i = m_i/\sum_{i=1}^{N} m_i$ and $N_{WS}$ is the number of Wigner–Seitz cells in the system[92,93]. Finally, the Gibbs free energy was determined as

$$G^{FL}(N, P, T) = F_E(N, \langle V \rangle, T) + \Delta F + \Delta F_{CM} + P\langle V \rangle, \quad (22)$$

where $\Delta F$ was calculated by Eq. (11) from the nonequilibrium switching[33].

**Free energy of vacancy.** We used $5 \times 5 \times 5$ supercell of BCC iron (250 atoms). The iron atom at the center of the supercell was removed to simulate the vacancy. We determined the spring constant from the NVT simulation of the perfect crystal and used the same spring constant for both Frenkel–Ladd paths of the perfect crystal and the crystal with vacancy and for the alchemical pathway of switching the atom into a spring. All NPT simulations were conducted under a pressure of 1 atm. Before the alchemical switching process, the initial system was equilibrated for 20 ps using the Berendsen thermostat and barostat (inhomogeneous) to reduce initial fluctuations in the cell volume. Then, the Nose–Hoover and Parrinello–Rahman dynamics[94] were used to simulate the switching process. The same $\lambda$ scheduling for the Frenkel–Ladd path was used, with equilibration and switching times of 40 ps and 60 ps, respectively. The alchemical free energy change $\Delta G^{AL}$ was determined using Eq. (11), and satisfy the following relationship:

$$\Delta G^{AL} = (G_{defect} + F_{spring}) - G_{perfect}, \quad (23)$$

where the free energy of spring $F_{spring}$ could be computed from Eq. (20) with $N = 1$.

**Alchemical free energy calculations.** We started from $6 \times 6 \times 6$ supercell of $\alpha$-CsPbI$_3$ for perovskite phase and $6 \times 3 \times 3$ supercell of $\delta$-CsPbI$_3$ for non-perovskite phase. Both systems contain 1080 atoms. For the alchemical pathway, we used the same simulation procedure as the alchemical pathway for the vacancy, under a pressure of 1 atm. Additionally, we set the masses of atoms as the weighted sum of masses of alchemical atoms, i.e., $m_i(\lambda) = (1-\lambda)m_i^{(i)} + \lambda m_i^{(f)}$, through the switching process. The alchemical free energy change was computed as

$$\Delta G^{AL} = \Delta G + \frac{3}{2}k_B T \sum_{i=1}^{N} \ln\left[\frac{m_i^{(i)}}{m_i^{(f)}}\right], \quad (24)$$

where $\Delta G$ is computed using Eq. (11). The second term on the right-hand side accounts for the change in masses over the transformation and originates from kinetic energy contributions[33].

**Reporting summary**

Further information on research design is available in the Nature Portfolio Reporting Summary linked to this article.

## Data availability

The initial structures used in this work are available in the Materials Project[16], with the material IDs of mp-20194 (CeO$_2$), mp-23324 (BiSBr), mp-22851 (NaCl), mp-804 (hexagonal GaN), mp-661 (hexagonal AlN), mp-13 (BCC Fe), mp-1069538 ($\alpha$-CsPbI$_3$), and mp-540839 ($\delta$-CsPbI$_3$). The processed dataset for perovskite oxide ordering, originally sourced from refs. 52,53, is available in the accompanying GitHub repository[95] referenced in the Code Availability section. The result files for the free energy calculations have been deposited in Zenodo under the accession code 11081396: https://doi.org/10.5281/zenodo.11081396[96]. Source data are provided with this paper.

## Code availability

The code to reproduce this work is publicly accessible on GitHub: https://github.com/learningmatter-mit/alchemical-mlip[95].

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

## Acknowledgements

The authors would like to thank Johannes C.B. Dietschreit for detailed feedback on the manuscript, and Xiaochen Du, Soojung Yang, Sulin Liu, Dadam Kang, Pablo Leon, Hoje Chun, Akshay Subramanian, and Nofit Segal for the helpful suggestions and discussions. We acknowledge the MIT SuperCloud and Lincoln Laboratory Supercomputing Center for providing HPC resources. J.N. was supported by the Toyota Research Institute and the Ronald A. Kurtz Fellowship. J.P. was supported by the Under Secretary of Defense for Research and Engineering under Air Force Contract No. FA8702-15-D-0001. Any opinions, findings, conclusions or recommendations expressed in this material are those of the author(s) and do not necessarily reflect the views of the Under Secretary of Defense for Research and Engineering.

## Author contributions

J.N. and R.G-B. conceived the project. J.N. designed the methodologies and developed the algorithms. J.N. and J.P. implemented the code and analyzed the results. R.G-B. supervised the research. All authors discussed the results and contributed to the final manuscript.

## Competing interests

The authors declare no competing interests.
