## [Transparent Peer Review file · Nature Communications]

Interpolation and differentiation of alchemical degrees of freedom in machine learning interatomic potentials

Corresponding Author: Professor Rafael Gomez-Bombarelli

Version 0:

Reviewer comments:

Reviewer #1

(Remarks to the Author)

In this work, the authors introduce a modification to a universal machine learning interatomic potential (MLIP) to facilitate alchemical transformations. Specifically, they have modified the message passing and readout of the MACE-MP-0 model to reflect atoms of mixed identities. The applicability of their model is exemplified by computing lattice constants of solid solutions as well as free energy differences of alchemical transformations.

The manuscript is well written and the results are presented in a clear and convincing manner. While the results are quite interesting, in particular, the compositional optimization of the solid solutions, they are not groundbreaking in a sense that they would revolutionize the field. In the Supplementary Information, the authors write themselves:

"Hence, the alchemical scheme developed in this work could be regarded as a computationally efficient alternative to naive linear interpolation of endpoint energies."

This is a fair assessment and while this is still an important contribution to the field, I recommend publication in a more specialized journal.

Some additional comments/remarks:

- p.4, left column: "We start from BiSBr structure (Fig. 2a) and split the Br atoms into three alchemical atoms of Cl, Br, and I." This is somewhat confusing as there is never an interpolation between all 3 species, instead one component of the λ vector is set to 0. Why did the authors chose this representation instead of simply using 2 components?

- p.5, Fig.2: the authors do comment on the discrepancies between the experimental and computed lattice constants of the pure phases; they should, however, still show the same ranges on the y-axes of the graphs. Specifically, the experimental and computed a and c lattice parameters of $\text{BiS}_{1-x}\text{Y}_x$ exhibit quite different ranges

- p.4, right column: "In this case, since the size of Na is between Li and K, multiple optimal compositions exist on the compositional space." It remains unclear what this means for the optimization. If there is no unique solution, how does the optimization converge?

- p.6, Fig.3a: it is unclear what the legends with the color gradients correspond to

- p.7, right column: the authors discuss that there might be a phase transformation from α to β occurring along the alchemical path; if this is indeed the case, the results for ΔG are not reliable; it might be fortunate that the free energy of the α and β phase are so similar that it does not matter much in this case, but it does not represent the ΔG in Eq.(13); also, it should be easy to detect from the analysis of the simulation data if this is the case and the authors should check this in detail

(Remarks on code availability)

Reviewer #2

(Remarks to the Author)

The authors present a generalisation of MLIP, illustrated on MACE, where they enable models to perform quantum alchemy calculations in solid state. I find the work to be well written and think it is an important contribution. However, if my understanding below is correct, a better discussion of the limitations of this approach is warranted.

While quite a bit of consideration went into recovering the correct limit behavior of the alchemical model to recover the regular models for the appropriate weights, I think it would be prudent to emphasize more that this is what is known as Virtual Crystal Approximation (VCA) in quantum chemistry, see e.g. DOI 10.1103/PhysRevB.89.165201, 10.1103/PhysRevB.61.7877, more references going back to the 1950s. In this work (in my understanding), the atom identity is interpolated by weighting subsets of networks s.t. for each interpolated site some (artificial) path is built. This way, the local ordering of the solid mixtures are ignored altogether, effectively inheriting all limitations of the (rarely used) VCA method, such as the requirement that elements be similar in volume, packing and location, and that disorder should not play a crucial role. Making this connection in the manuscript might be quite helpful for the reader and connects to a wide body of literature in quantum chemistry.

In quantum alchemy, $\partial E / \partial \lambda$ is often accessible from quantum chemistry calculations. While such calculations sometimes face the same issues the authors discuss for the embedding interpolation (e.g. when interpolating PAW potentials or pseudopotentials, see 10.1002/aic.17041, 10.1063/5.0079483), analytical gradients are available (e.g. 10.1063/5.0076202, 10.1021/acscentsci.7b00586, 10.1063/5.0118200). Would it help to include that information during training or would the asymmetric definition exclude the use of physical derivatives?

Eq 8 (and following ones) requires the integrand to be a state function. In this approach, where the interpolation path is artificial (meaning unconnected to physical interpolation paths by e.g. varying Z_i in quantum chemistry calculations), how is the well-definedness of the integral guaranteed? E.g. if we were to change the number of electrons along such a path, the MLIP would happily learn this despite the non-differentiability of the energy w.r.t. number of electrons - i.e. provide an unphysical potential.

I think this is a valuable body of work which would benefit from a) a more extensive discussion of limitations and b) a more explicit connection to the quantum chemistry.

(Remarks on code availability)

Version 1:

Reviewer comments:

Reviewer #1

(Remarks to the Author)

I appreciate that the authors have extended their manuscript to highlight the applicability and usefulness of their approach in particular regarding the description of disordered systems. In this context, a comparison with SQS and VCA (as pointed out by the other reviewer) is most suitable and this has been addressed appropriately in the revised manuscript. The additional application to perovskite oxides is quite interesting, though it seems that performance regarding classification of ordered/disordered structure does not outperform the SQS estimate which is attributed to local structural distortions which cannot be captured by the current approach.

In terms of novelty and impact, the free energy calculations appear less impressive, as already discussed in the first round of reviews. This part by itself, I would not evaluate as significant enough to warrant publication in Nature Communications. The application to disordered solid solutions, in particular on the basis of a pre-trained MLIP, could potentially be more impactful.

Regarding the free energy calculations and the comparison with the Frenkel-Ladd path, there still seems to be a slight discrepancy in what is compared in Fig.6b. I understand the authors' explanation where this difference might come from in terms of changes in structure and volume and the updated nomenclature make it much more clear. My concern is if this is a valid or the most useful comparison between the Frenkel-Ladd and alchemical calculations. The corresponding free energy equations are discussed in Sec. IVC, in particular Eq.(20) to convert the free energy F at constant V to the Gibbs free energy G at constant P . This makes sense if the two endpoints (in this case the Einstein crystal and the 'real' system) have the same equilibrium volume at a given pressure and temperature (or, to be more precise, there is no pressure in the Einstein crystal; thus, the simulation is performed at the equilibrium V of the 'real' system). When computing $\Delta G(\text{Pb} \rightarrow \text{Sn})$ for the perovskite phase using the Frenkel-Ladd path, there are two options:

(a) apply Eq.(8) where the initial and final Hamiltonian are the two perovskite phases CsPbI_3 and CsSnI_3 at the same volume and in the same structure (α) to directly obtain ΔF

(b) perform two separate simulations between the Einstein crystal and CsPbI_3 and CsSnI_3 at their respective equilibrium volumes (and shapes, so α for CsSnI_3 and β for CsPbI_3) to obtain the absolute free energies of the two perovskite phases and then compute the difference from there

It seems as if (a) was used in the current manuscript? And if this is the case, then this would not be the correct ΔG or at least not a ΔG at constant P since the two chemically different perovskite phases appear to have different volumes and different cell shapes? If (b) was applied, then it should give the same as the alchemical path since at lower T, the β phase of CsPbI₃ should have been the corresponding equilibrium structure to consider for the Frenkel-Ladd path. Would be great if the authors can clarify this.

(Remarks on code availability)

Reviewer #3

(Remarks to the Author)

The introduction is mostly written in a way that - while technical - is accessible to a broad scientific audience with a basic background in materials science or machine learning. The authors could try to make the last part of the introduction even more accessible to a broader audience by thinking how to simplify the sentences "Rather than altering the continuous embeddings of individual atoms, we augment the input graph structure by introducing alchemical atoms, each associated with its respective compositional weight. Through subsequent modifications to the message passing scheme and energy readout, our scheme provides smooth interpolation between different compositional states of materials." in a more descriptive way, maybe giving an example. In the results section, The authors take care to introduce MLIPs and then motivate the idea of "alchemical" degrees of freedom before diving into technical details. Some parts remain specialized, but overall the exposition (especially in the introduction and first results) makes the study's context understandable.

The goal is stated clearly: to modify universal MLIPs (in particular, the MACE-MP-0 model) so that they can continuously interpolate between different compositional states. This enables efficient compositional optimization, accurate free energy calculations (e.g., vacancy formation), and improved disorder modeling in complex materials.

The paper makes it clear that by introducing alchemical weights into the graph representation and modifying the message passing/readout steps, the model recovers the original behavior in limiting cases while also extending its capabilities. The new method is implemented as an extension of MACE-MP-0. Throughout the results, the "alchemically modified MACE-MP-0" is used for various tasks (e.g., predicting lattice parameters of solid solutions, free energy calculations, and disorder energetics).

While MACE-MP-0 already models different compounds over a large chemical space, the new approach goes further by allowing continuous interpolation between compositional states (which traditional discrete representations do not offer). It also enables efficient gradient-based optimization of compositions, which is particularly beneficial for free energy and disorder calculations. Furthermore, the approach reduces computational cost by enabling smaller alchemical supercells that still capture essential disorder physics (as shown in comparisons with SQS methods). These advantages demonstrate that the alchemical extension not only preserves the accuracy of the baseline model in known limits but also provides a practical pathway for exploring compositional changes and disorder with greater efficiency.

The rebuttal letter shows that the authors have addressed several points raised by reviewers. They clarify ambiguities (such as the interpolation representation and figure legends), add extra experimental results (e.g., additional data on disorder energetics and phase stability), and contextualize their approach with respect to existing methods like the Virtual Crystal Approximation. In my opinion, the concerns have been sufficiently addressed through the authors' responses and the incorporation of revised figures and text.

(Remarks on code availability)

There is a readme and jupyter notebook files that simplify getting started. I didn't have the time or computational resources to thoroughly check the code and run it.

Response to Reviewer #1

In this work, the authors introduce a modification to a universal machine learning interatomic potential (MLIP) to facilitate alchemical transformations. Specifically, they have modified the message passing and readout of the MACE-MP-0 model to reflect atoms of mixed identities. The applicability of their model is exemplified by computing lattice constants of solid solutions as well as free energy differences of alchemical transformations.

The manuscript is well written and the results are presented in a clear and convincing manner. While the results are quite interesting, in particular, the compositional optimization of the solid solutions, they are not groundbreaking in a sense that they would revolutionize the field. In the Supplementary Information, the authors write themselves: “Hence, the alchemical scheme developed in this work could be regarded as a computationally efficient alternative to naive linear interpolation of endpoint energies.” This is a fair assessment and while this is still an important contribution to the field, I recommend publication in a more specialized journal.

We appreciate the reviewer’s acknowledgment of the clarity and significance of our results, as well as their thoughtful suggestions for addressing ambiguities in our exposition. We would like to clarify that the statement in the Supplementary Information (SI) highlighted by the reviewer specifically pertains to the discussion of free energy calculations, where correct results are theoretically guaranteed as long as the endpoints are accurate. We would like to respectfully emphasize that this statement highlighted by the reviewer does not apply beyond free energy calculations, such as for the efficient and accurate modeling of disorder materials with partial elemental site occupancy. We believe that our approach could be particularly valuable in situations where the compositional complexity is high enough that evaluating all possible “endpoints” becomes infeasible, offering a practical and efficient alternative for such challenging cases.

To further elaborate our points, we conducted a new set of experiments on disordered solid solutions of the multicomponent perovskite oxide $A_2B'B''O_6$, with the dataset obtained from our recent works [1, 2]. In summary, we demonstrate that alchemical modification of MLIPs offers an efficient and scalable approach for modeling cation disorder in such systems, achieving reasonable accuracy in predicting experimental order/disorder when compared to larger special quasirandom structures (SQS), but with significantly reduced computational cost. We have added an extra section at the end of Sec. II B and Fig. 4 in the main text, as reproduced below.

We hope these new results further strengthen the novelty and significance of our work by emphasizing that the alchemical scheme is not merely a replacement for linear energy interpolation, but a meaningful and efficient surrogate for modeling disorder in the context of MLIPs.

- **Sec. II B, disordered energetics and Fig. 4 (added)**

Disorder energetics. The high computational efficiency and accuracy of alchemically modified MLIPs for modeling disordered solid solutions are further validated by examining a dataset of $A_2B'B''O_6$ multicomponent perovskites in our recent high-throughput studies [1, 2]. Notably, the thermodynamic preference of an $A_2B'B''O_6$ perovskite to adopt either cation-ordered or cation-disordered structures depends on the difference between formation energetics of various cation-ordered configurations and those of cation-disordered solid solutions [2]. For ordered structures, the formation energetics across all possible symmetrically inequivalent cation arrangements can serve as physics-informed descriptors to predict the

Figure 4: **Disordered energetics in multicomponent perovskite oxides.** (a) Crystal structure schematics for fully cation-disordered $A_2B'B''O_6$ perovskite oxide solid solutions, illustrating different alchemical supercell sizes and number of atoms, and representative 320-atom $4 \times 4 \times 4$ SQS structures. (b) MACE-relaxed energy differences between cation-disordered $2 \times 2 \times 2$ alchemical cells and smaller or larger supercells, evaluated across 100 $A_2B'B''O_6$ compositions from [2]. (c) Difference between the unrelaxed and MACE-relaxed structures for various alchemical cell sizes, including $4 \times 4 \times 4$ SQS structures, quantified by cosine distance between local structure fingerprints [1, 3]. (d) Comparison of MACE-relaxed energies for $2 \times 2 \times 2$ alchemical cells versus $4 \times 4 \times 4$ SQS structures. (e) MACE-relaxed energy differences among $2 \times 2 \times 2$ alchemical cells, $4 \times 4 \times 4$ SQS structures, and all cation-ordered configurations with four B' and four B'' cations on eight B sites in the $2 \times 2 \times 2$ supercell. Compositions are sorted by energy differences between $4 \times 4 \times 4$ SQS and lowest-energy ordered arrangements. Experimentally characterized ordered and disordered compositions [1] are marked in the upper and lower regions, respectively. (f) Receiver operating characteristic (ROC) curves for experimental order/disorder classification [1] based on relative energy values of $4 \times 4 \times 4$ SQS or $2 \times 2 \times 2$ alchemical cells in (e) with respect to the lowest-energy cation-ordered arrangements, with area under the curve (AUC) values shown. All results are derived using the alchemically modified MACE-MP-0 medium model [4].

thermodynamic tendency towards experimental cation disorder [1]. While DFT is computa-

tionally prohibitive for evaluating formation energetics of various enumerated cation-ordered atomic arrangements, we have shown that symmetry-aware equivariant graph neural networks, including equivariant MLIPs, provide efficient and accurate surrogates for assessing ordering-dependent thermodynamic stability in multicomponent perovskite oxides [2].

In this work, we extend our previous analysis to directly examine the formation energetics of fully cation-disordered $A_2B'B''O_6$ solid solutions with partial B site occupancies of 0.5 B' and 0.5 B'' . Traditionally, special quasirandom structure (SQS, [5, 6]), which optimizes elemental placements within a supercell to mimic the cluster vectors of random alloys, has been widely used to study disordered solid solutions. While SQS provides a systematic approach to model disordered structures, it requires large supercells to avoid correlations across periodic boundaries and relies on optimization routines such as Monte Carlo simulations [7], limiting its feasibility for high-throughput studies. Given the efficiency of alchemically modified MLIPs in representing disorder through partial elemental occupancies, we compare alchemical unit cell modeling of perovskite solid solutions to SQS cells using baseline MLIPs for disorder modeling.

Starting from the base ordered perovskite ABO_3 structure, we split the B atom into two alchemical species, B' and B'' , each assigned an alchemical weight of 0.5. We then generate $N \times N \times N$ ($N = 1, 2, 4, 6$) alchemical supercells and $4 \times 4 \times 4$ SQS supercells (Fig. 4a), optimizing each cell using alchemically modified MACE-MP-0 and baseline MACE-MP-0 models. For alchemical supercells, the relaxed cell energy per atom plateaus at the $2 \times 2 \times 2$ supercell, while the unit cell ($1 \times 1 \times 1$) exhibits notably higher energy compared to larger supercells (Fig. 4b). The structural differences between unrelaxed and relaxed disordered cells, shown in Fig. 4c, quantified using cosine distances of local structural fingerprints, as done in [1, 3], reveal that the alchemical unit cell relaxes only slightly, whereas larger alchemical supercells and SQS cells show more significant differences between their corresponding unrelaxed and MLIP-relaxed structures. As noted in our previous works [1, 2], crystallographic sites undergo substantial distortion during relaxation, such as octahedral tilting and Jahn–Teller distortions [8], which are typically beyond the periodicity of a perovskite unit cell and thus can hardly be captured by modeling a single unit cell. Given that the $2 \times 2 \times 2$ supercell yields results similar to those of larger alchemical supercells, we proceed with further analysis using the 40-atom $2 \times 2 \times 2$ alchemical supercell.

As shown in Fig. 4d, the optimized single-point energies from the alchemical $2 \times 2 \times 2$ supercell align well with those from the $4 \times 4 \times 4$ SQS supercell, with a mean absolute error (MAE) of 0.032 eV/atom. Since the preference for cation-ordered and cation-disordered configurations depends on the relative formation energetics of each, we further compare energy values with all symmetrically inequivalent cation-ordered configurations in the $2 \times 2 \times 2$ supercell, obtained by enumerating four B' and four B'' cations occupying eight B sites [1]. The results in Fig. 4e show the relative energies of all considered structures, aligned with the ground-state (lowest-energy) cation-ordered structure energy as the reference. As previously discussed in [1, 2], we observe that experimentally observed ordered compositions exhibit significant difference between the ground-state ordered configuration energy and other configurations, whereas experimentally cation-disordered compositions show similar energies among different configurations. The relative energy of the disordered SQS supercell provides a useful metric for characterizing experimental order/disorder, as seen by the separation of ordered and disordered compositions when sorting the oxide compositions by the SQS energy. Although the $2 \times 2 \times 2$ alchemical supercell energies show more stochasticity, they follow the same trend as the relative energies of the SQS. This is further supported by the receiver operating characteristic (ROC) curves for

experimental order/disorder classification based on relative energy values (Fig. 4f) following our previous work [1], where $4 \times 4 \times 4$ SQS cell energies provide excellent classification with an area under the curve (AUC) of 0.95, while the alchemical $2 \times 2 \times 2$ cell energies achieve reasonably good experimental order/disorder classification with an AUC of 0.80. The likely source of this difference is that for ions of very different sizes, local structural distortions are related to local chemical ordering, but the use of an average structure imposed by the alchemical method fails to produce local distortions that SQS captures well.

Hence, based on these results, we conclude that the alchemical modification of MLIPs offers a scalable approach for disorder modeling, as demonstrated with this multicomponent perovskite oxide dataset. The alchemical $2 \times 2 \times 2$ supercells provide reasonable accuracy for experimental disorder classification, while using only 1/8 of the atoms in the $4 \times 4 \times 4$ SQS supercells. Unlike SQS, these alchemical supercells can be obtained without the need for additional annealing steps for configuration generation. The results were achieved by modifying off-the-shelf pre-trained MLIPs and could be further fine-tuned to improve energy prediction and order-disorder classification. They may also be adapted for other material systems, including compositionally complex alloys and ceramics.

Some additional comments/remarks:

- p.4, left column: “We start from BiSBr structure (Fig. 2a) and split the Br atoms into three alchemical atoms of Cl, Br, and I.” This is somewhat confusing as there is never an interpolation between all 3 species, instead one component of the λ vector is set to 0. Why did the authors chose this representation instead of simply using 2 components?

We agree with the reviewer that describing the method as an interpolation of two components provides a clearer explanation, and we have revised both the manuscript and the code accordingly. We would like to note that this change does not affect the results, as the two approaches are essentially equivalent, since zero weights do not contribute to the energy or gradient computations.

- **p. 4, left column (updated)**

We start from BiSBr structure (Fig. 2a) and split the Br atoms into ~~three alchemical atoms of Cl, Br, and I~~ two alchemical atoms of X and Y. The cell parameters are optimized with respective alchemical weights. For example, the BiS $\text{Cl}_{1-x}\text{I}_x$ structure will have alchemical atoms Cl and I with alchemical weights of $\lambda = (1-x, x)$ ~~$\lambda = (1-x, 0, x)$~~ .

- p.5, Fig.2: the authors do comment on the discrepancies between the experimental and computed lattice constants of the pure phases; they should, however, still show the same ranges on the y-axes of the graphs. Specifically, the experimental and computed a and c lattice parameters of BiS X_{1-x}Y_x exhibit quite different ranges

We have updated the Fig. 2 (and Fig. S3) to ensure that the experimental and computed lattice parameters are presented with consistent y-axis ranges for better comparison.

- **Fig. 2 (updated, see next page)**

- p.4, right column: “In this case, since the size of Na is between Li and K, multiple optimal compositions exist on the compositional space.” It remains unclear what this means for the optimization. If there is no unique solution, how does the optimization

Figure 2: **Lattice parameters for solid solutions.** (a) The starting structures, CeO_2 and BiSBr , for solid solutions. (b) Lattice parameter a for $\text{Ce}_{1-x}\text{M}_x\text{O}_2$ ($M = \text{Zr}, \text{Sn}$) as a function of the compositional weight x . (c) Lattice parameters a , b , and c for $\text{BiSX}_{1-x}\text{Y}_x$ ($X, Y = \text{Cl}, \text{Br}, \text{I}$) as a function of x . The upper panels are the result of the alchemically modified MACE-MP-0 medium model [4], and the lower panels are the experimental results from [9] and [10] for (b) and (c), respectively. Arrows in the rightmost panels indicate the composition with the minimum value of \$c\$.

converge?

We would like to note that numerical optimization can still converge, even in the absence of a unique point of optimality. In cases where multiple optimal solutions exist, the gradient along the line (or curve) composed of these optimal values is zero. As a result, the optimization process will converge when it reaches any point along this line or curve of optimal solutions.

- p.6, Fig.3a: it is unclear what the legends with the color gradients correspond to

To improve clarity, we have revised the caption to explain the meaning of the color gradients in each panel and added dotted lines to each plot to indicate the optimized composition line (addressing a related point from the previous comment as well).

- **Fig. 3 (updated, see next page)**

- p.7, right column: the authors discuss that there might be a phase transformation from α to β occurring along the alchemical path; if this is indeed the case, the results for ΔG are not reliable; it might be fortunate that the free energy of the α and β phase are so similar that it does not matter much in this case, but it does not represent the ΔG in Eq.(13); also, it should be easy to detect from the analysis of the simulation data if this is the case and the authors should check this in detail

We appreciate the reviewer for their insightful comment. In response, we have revised the main text to use “P” (perovskite) and “N” (non-perovskite) in place of α and δ to improve clarity regarding phase transitions. Additionally, we have added a new section in the Supplementary

Figure 3: **Compositional optimization.** (a) Lattice parameter optimization for solid solutions of LiCl, NaCl, and KCl. The left panel shows the lattice parameters optimized lattice parameters as a color gradient, obtained by relaxing the cell geometry at for each compositional weight, and the right panel shows the absolute hydrostatic stress values obtained. The right panel displays hydrostatic stress, with color intensity representing stress magnitude and arrows indicating gradient direction, calculated by fixing the cell dimension to that dimensions to those of NaCl. Because Since the energy output is end-to-end differentiable with respect to the alchemical weights, the gradient of the stress with respect to the composition could be computed (black arrows in the right panel), which enables the optimization of the composition to match the given cell dimensions composition can be optimized to match target cell dimensions (minimizing stress) by following these gradients. The compositions with cell parameters matching NaCl (left) and those obtained by minimizing stress (right), indicated by the dotted lines, align in both figures. (b) The optimization for the lattice-matching condition for solid solutions $\text{Al}_{1-x}\text{Sc}_x\text{N}$ and $\text{Al}_{1-x}\text{Y}_x\text{N}$ with GaN. The most stable polymorph structures are shown on the left. The plot on the right shows the cell dimension a obtained by optimizing for each compositional weight (Scan), calculated from the corresponding supercell (Supercell), and the compositional weights optimized by gradient descent to match the a value for GaN (Optimized). All results are obtained using the alchemically modified MACE-MP-0 medium model [4].

Information (Sec. III) to address the reviewer’s concern about the free energy difference along the alchemical path. We hope this addition clarifies the reliability of our ΔG calculations.

- **SI, Sec. III (added)**

Here, we examine the impact of phase transformation on the discrepancy between the Frenkel-Ladd and alchemical paths, as shown in Fig. 6 of the main text. Each free energy simulation begins from the energy-minimized structure of $\alpha\text{-CsPbI}_3$. In the Frenkel-Ladd path (NVT), we equilibrate the system using a homogeneous Berendsen thermostat, which maintains the relative scales of the cell parameters, while the alchemical path (NPT)

Figure 4: **Equilibrium cell parameters and free energies for perovskite phases.** (Left) Evolution of cell parameters a , b , and c for perovskite CsPbI_3 structures at 500 K and 350 K during NPT equilibration. (Right) Schematic of corresponding free energy profiles at each temperature. Equilibrium cell parameters of CsPbI_3 reveal the lowest free energy state between perovskite phases α and β , highlighting differences between free energies calculated via the Frenkel–Ladd path (NVT) and the alchemical path (NPT).

is equilibrated using an inhomogeneous Berendsen thermostat. We track the evolution of cell parameters during alchemical path equilibration in Fig. 4 (left) at 500 K and 350 K. At 500 K, all cell parameters remain consistent (cubic, α), while at 350 K, they equilibrated into a tetragonal structure with $a \approx c < b$, indicating an initial transition from α to β . Since the structure at CsSnI_3 composition remains in the α (cubic) phase, as shown in the free energy schematic (Fig. 4, right), the lower free energy perovskite phase (β) of CsPbI_3 leads to a larger free energy difference for the alchemical path. Since our definition of “phase” in the main text is based on the highest free energy barrier—the transition between perovskite (P) and non-perovskite (N) structures—the alchemical path simulated in the NPT ensemble, which accounts for the lower free energy perovskite phase (α or β) at each temperature, aligns better with our goal of estimating free energy differences.

Response to Reviewer #2

The authors present a generalisation of MLIP, illustrated on MACE, where they enable models to perform quantum alchemy calculations in solid state. I find the work to be well written and think it is an important contribution. However, if my understanding below is correct, a better discussion of the limitations of this approach is warranted.

We appreciate the reviewer for recognizing the significance of our work and for suggesting valuable literature to help contextualize it within the field of quantum alchemical models. We have addressed each point regarding the limitations of our approach to provide greater clarity in our responses to the subsequent comments.

While quite a bit of consideration went into recovering the correct limit behavior of the alchemical model to recover the regular models for the appropriate weights, I think it would be prudent to emphasize more that this is what is known as Virtual Crystal Approximation (VCA) in quantum chemistry, see e.g. DOI 10.1103/PhysRevB.89.165201, 10.1103/PhysRevB.61.7877, more references going back to the 1950s. In this work (in my understanding), the atom identity is interpolated by weighting subsets of networks s.t. for each interpolated site some (artificial) path is built. This way, the local ordering of the solid mixtures are ignored altogether, effectively inheriting all limitations of the (rarely used) VCA method, such as the requirement that elements be similar in volume, packing and location, and that disorder should not play a crucial role. Making this connection in the manuscript might be quite helpful for the reader and connects to a wide body of literature in quantum chemistry.

We appreciate the reviewer’s insightful observation regarding the connection between our method and the Virtual Crystal Approximation (VCA). Indeed, our approach shares important similarities with VCA. To clarify, VCA rests on two key assumptions: (1) *geometry*: the solid solution is modeled by an averaged structure where the same crystallographic sites are randomly occupied by different elements, thereby neglecting local ordering, and (2) *interaction*: the random occupancy is approximated by compositional weighting of the atomic pseudopotentials.

Our method adopts the first assumption, so the limitations highlighted by the reviewer concerning geometry apply to our approach as well: the elements should have similar sizes, occupy analogous positions, and the effects of local disorder should not be significant.

However, the limitations of VCA may also stem from “pseudopotential alchemy,” where the accuracy of results depends heavily on carefully tuning pseudopotential parameters such as radial cutoffs and electronic configurations. In contrast, our method is largely free from these challenges, as MLIPs replace the need for detailed electronic structure calculations with repeated message-passing between node and edge features. The regularization from training scheme and model architecture helps ensure that results remain within a reasonable physical range, reducing the need for extensive manual adjustment of calculation parameters.

Therefore, we cautiously conclude that while our method inherits the geometrical requirements of VCA, it significantly reduces the need for manual tuning of computational parameters to achieve reasonable results. We have incorporated this discussion into the revised manuscript in the following sections.

- **Sec. II B, final paragraph (added)**

We also note that our approach shares similarities with the Virtual Crystal Approximation

(VCA, [11, 12, 13]), a traditional approach in modeling solid solutions with partial elemental site occupancy. VCA relies on two assumptions: (1) *geometry*: the solid solution is represented by an averaged structure where crystallographic sites are randomly occupied by different elements, disregarding local ordering; and (2) *interaction*: the random occupancy is approximated by compositionally weighted average of atomic pseudopotentials. Our method adopts the first assumption, making it subject to the same geometric limitations, such as the elements should be of similar size, occupy comparable positions, and local disorder effects should be minimal. However, the practical limitations of VCA mainly arise from what could be described as “pseudopotential alchemy,” where accuracy depends heavily on carefully tuning pseudopotential parameters like radial cutoffs and electronic configurations (core/valence). In contrast, our method sidesteps these challenges: MLIPs replace electronic structure calculations with iterative message-passing between node and edge features. Built-in regularization from training scheme and model architecture help ensure that results remain within a physically reasonable range, reducing the need for extensive manual parameter adjustments.

In quantum alchemy, $\partial E/\partial\lambda$ is often accessible from quantum chemistry calculations. While such calculations sometimes face the same issues the authors discuss for the embedding interpolation (e.g. when interpolating PAW potentials or pseudopotentials, see 10.1002/aic.17041, 10.1063/5.0079483), analytical gradients are available (e.g. 10.1063/5.0076202, 10.1021/acscentsci.7b00586, 10.1063/5.0118200). Would it help to include that information during training or would the asymmetric definition exclude the use of physical derivatives?

We thank the reviewer for their insightful suggestion. As noted, our current alchemical scheme does not alter atomic identities directly. Instead, it controls the weights of alchemical atoms with fixed elemental identities. In this setup, λ represents the occupancies of different atomic types rather than coupling directly to atomic numbers. This design aligns with our intention to leverage pre-trained MLIPs for our application.

However, while pre-trained embeddings may not be ideal, one could fine-tune these embeddings to align with the “alchemical force” $\partial E/\partial\lambda$ derived from quantum alchemy, possibly using analytical gradients, given that the baseline MLIP is end-to-end differentiable with respect to embeddings. Although this adjustment could enhance the physical relevance of alchemical degrees of freedom, it is incompatible with the current scheme, so we leave it as a prospect for future work. We have also addressed this point in the Discussion section of the main text, along with the following reviewer comment (see below).

Eq 8 (and following ones) requires the integrand to be a state function. In this approach, where the interpolation path is artificial (meaning unconnected to physical interpolation paths by e.g. varying Z_i in quantum chemistry calculations), how is the well-definedness of the integral guaranteed? E.g. if we were to change the number of electrons along such a path, the MLIP would happily learn this despite the non-differentiability of the energy w.r.t. number of electrons - i.e. provide an unphysical potential.

Indeed, the thermodynamic integration (TI) integrand should represent a state function. However, we note that in the context of free energy calculations in this work, the interpolation state parametrized by λ is not intended to map onto a physically realizable state (e.g., corresponding to specific atomic numbers Z_i). While quantum alchemy constructs a path of physically realizable states between two endpoints and approximates them using methods like density functional

theory (DFT), the TI path used here involves interpolation between approximations—in this case, Hamiltonians parametrized by neural networks.

Thus, the intermediate states are virtual, in the sense that they do not correspond to actual physical states. However, their well-definedness is ensured because they represent thermal ensembles governed by an interpolated neural network Hamiltonian with the alchemical parameter λ . As long as the endpoint states are good approximations of physical reality and the energy interpolation remains sufficiently smooth, the free energy calculated via TI will converge to a physically meaningful value. Nevertheless, as noted in the previous point, learning an MLIP-based representation consistent with quantum alchemy (varying Z_i) might be able to offer a well-regularized approximation of the physical state for the TI calculation. We will explore this further as we refine the approach. We have updated the Discussion section in the main text as follows:

- **Sec. III, final paragraph (added)**

While we devised the alchemical scheme with fixed elemental identities and λ representing the occupancies of different alchemical atoms to align with our goal of leveraging pre-trained MLIPs, we note that interpolating the elemental identities of atoms, i.e., coupling λ to atomic numbers, is also a promising direction that connects with the quantum alchemy literature. Although pre-trained embeddings may not be ideal for this, they could be fine-tuned by “alchemical force matching” with $\partial E/\partial\lambda$ derived from quantum alchemy [14, 15, 16], possibly using analytical gradients [17, 18, 19], given that the baseline MLIP is end-to-end differentiable with respect to embeddings. Learning an MLIP-based representation consistent with quantum alchemy might offer a well-regularized approximation of the physical state for the TI calculation. While this alternative scheme could improve the physical relevance of alchemical degrees of freedom, it is incompatible with the current approach and remains a prospect for future work.

I think this is a valuable body of work which would benefit from a) a more extensive discussion of limitations and b) a more explicit connection to the quantum chemistry.

We thank the reviewer again for their constructive feedback on our work. We hope that our replies and revisions have addressed the concerns regarding the limitations of our approach and the connection to quantum chemistry.

References

- [1] J. Peng, J. Damewood, and R. Gómez-Bombarelli, *Cell Rep. Phys. Sci.* **5**, 101942 (2024).
- [2] J. Peng, J. Damewood, J. Karaguesian, J. R. Lunger, and R. Gómez-Bombarelli, “Learning ordering in crystalline materials with symmetry-aware graph neural networks,” (2024), arXiv:2409.13851 [cond-mat.mtrl-sci] .
- [3] J. N. Law, S. Pandey, P. Gorai, and P. C. St. John, *JACS Au* **3**, 113 (2023).
- [4] I. Batatia, P. Benner, Y. Chiang, A. M. Elena, D. P. Kovács, J. Riebesell, X. R. Advincula, M. Asta, M. Avaylon, W. J. Baldwin, F. Berger, N. Bernstein, A. Bhowmik, S. M. Blau, V. Cărare, J. P. Darby, S. De, F. D. Pia, V. L. Deringer, R. Elijošius, Z. El-Machachi, F. Falcioni, E. Fako, A. C. Ferrari, A. Genreith-Schriever, J. George, R. E. A. Goodall, C. P. Grey, P. Grigorev, S. Han, W. Handley, H. H. Heenen, K. Hermansson, C. Holm, J. Jaafar, S. Hofmann, K. S. Jakob, H. Jung, V. Kapil, A. D. Kaplan, N. Karimitari, J. R. Kermode, N. Kroupa, J. Kullgren, M. C. Kuner, D. Kuryla, G. Liepuoniute, J. T. Margraf, I.-B. Magdău, A. Michaelides, J. H. Moore, A. A. Naik, S. P. Niblett, S. W. Norwood, N. O’Neill, C. Ortner, K. A. Persson, K. Reuter, A. S. Rosen, L. L. Schaaf, C. Schran, B. X. Shi, E. Sivonxay, T. K. Stenczel, V. Svahn, C. Sutton, T. D. Swinburne, J. Tilly, C. van der Oord, E. Varga-Umbrich, T. Vegge, M. Vondrák, Y. Wang, W. C. Witt, F. Zills, and G. Csányi, “A foundation model for atomistic materials chemistry,” (2024), arXiv:2401.00096 [physics.chem-ph] .
- [5] S.-H. Wei, L. G. Ferreira, J. E. Bernard, and A. Zunger, *Phys. Rev. B* **42**, 9622 (1990).
- [6] A. Zunger, S.-H. Wei, L. G. Ferreira, and J. E. Bernard, *Phys. Rev. Lett.* **65**, 353 (1990).
- [7] A. van de Walle, P. Tiwary, M. de Jong, D. Olmsted, M. Asta, A. Dick, D. Shin, Y. Wang, L.-Q. Chen, and Z.-K. Liu, *Calphad* **42**, 13 (2013).
- [8] G. King and P. M. Woodward, *J. Mater. Chem.* **20**, 5785 (2010).
- [9] T. Baidya, P. Bera, O. Kröcher, O. Safonova, P. M. Abdala, B. Gerke, R. Pöttgen, K. R. Priolkar, and T. K. Mandal, *Phys. Chem. Chem. Phys.* **18**, 13974 (2016).
- [10] P. Schultz and E. Keller, *Acta Cryst. B* **70**, 372 (2014).
- [11] L. Nordheim, *Annalen der Physik* **401**, 607 (1931).
- [12] L. Bellaiche and D. Vanderbilt, *Phys. Rev. B* **61**, 7877 (2000).
- [13] C. Eckhardt, K. Hummer, and G. Kresse, *Phys. Rev. B* **89**, 165201 (2014).
- [14] C. D. Griego, L. Zhao, K. Saravanan, and J. A. Keith, *AIChE J.* **66**, e17041 (2020).
- [15] B. Huang and O. A. von Lilienfeld, *Chem. Rev.* **121**, 10001 (2021).
- [16] E. A. Eikey, A. M. Maldonado, C. D. Griego, G. F. von Rudorff, and J. A. Keith, *J. Chem. Phys.* **156**, 064106 (2022).
- [17] T. Tamayo-Mendoza, C. Kreisbeck, R. Lindh, and A. Aspuru-Guzik, *ACS Cent. Sci.* **4**, 559 (2018).
- [18] M. F. Kasim, S. Lehtola, and S. M. Vinko, *J. Chem. Phys.* **156**, 084801 (2022).
- [19] X. Zhang and G. K. Chan, *J. Chem. Phys.* **157**, 204801 (2022).

Response to Reviewer #1

I appreciate that the authors have extended their manuscript to highlight the applicability and usefulness of their approach in particular regarding the description of disordered systems. In this context, a comparison with SQS and VCA (as pointed out by the other reviewer) is most suitable and this has been addressed appropriately in the revised manuscript. The additional application to perovskite oxides is quite interesting, though it seems that performance regarding classification of ordered/disordered structure does not outperform the SQS estimate which is attributed to local structural distortions which cannot be captured by the current approach.

In terms of novelty and impact, the free energy calculations appear less impressive, as already discussed in the first round of reviews. This part by itself, I would not evaluate as significant enough to warrant publication in Nature Communications. The application to disordered solid solutions, in particular on the basis of a pre-trained MLIP, could potentially be more impactful.

We appreciate the reviewer's perspective on the novelty and impact and we are glad to hear that the additional experiments in the revision were well received.

Regarding the free energy calculations and the comparison with the Frenkel-Ladd path, there still seems to be a slight discrepancy in what is compared in Fig.6b. I understand the authors' explanation where this difference might come from in terms of changes in structure and volume and the updated nomenclature make it much more clear. My concern is if this is a valid or the most useful comparison between the Frenkel-Ladd and alchemical calculations. The corresponding free energy equations are discussed in Sec. IVC, in particular Eq.(20) to convert the free energy F at constant V to the Gibbs free energy G at constant P . This makes sense if the two endpoints (in this case the Einstein crystal and the 'real' system) have the same equilibrium volume at a given pressure and temperature (or, to be more precise, there is no pressure in the Einstein crystal; thus, the simulation is performed at the equilibrium V of the 'real' system). When computing $\Delta G(\text{Pb} \rightarrow \text{Sn})$ for the perovskite phase using the Frenkel-Ladd path, there are two options:

(a) apply Eq.(8) where the initial and final Hamiltonian are the two perovskite phases CsPbI_3 and CsSnI_3 at the same volume and in the same structure (α) to directly obtain ΔF

(b) perform two separate simulations between the Einstein crystal and CsPbI_3 and CsSnI_3 at their respective equilibrium volumes (and shapes, so α for CsSnI_3 and β for CsPbI_3) to obtain the absolute free energies of the two perovskite phases and then compute the difference from there

It seems as if (a) was used in the current manuscript? And if this is the case, then this would not be the correct ΔG or at least not a ΔG at constant P since the two chemically different perovskite phases appear to have different volumes and different cell shapes? If (b) was applied, then it should give the same as the alchemical path since at lower T , the β phase of CsPbI_3 should have been the corresponding equilibrium structure to consider for the Frenkel-Ladd path. Would be great if the authors can clarify this.

We thank the reviewer for their detailed comment. Between the two options, our approach aligns

more closely with (b), as we performed separate equilibrations and Frenkel–Ladd switching for each composition. However, since all simulations were initiated from the α phases and the Frenkel–Ladd path requires equilibrium atomic positions to which springs are attached, we employed a *homogeneous* Berendsen barostat to determine the equilibrium volume while preserving the original (α) structure shape. This part was missing in the Methods section, so we have added the following:

- **Main text Sec. IV C, page 12**

First, the system was equilibrated for 60 ps under the NPT ensemble with the pressure of $P = 1$ atm by the Berendsen thermostat and homogeneous Berendsen barostat.

Thus, even if the α phase is not the most stable at the given temperature, Eq. (20) (Eq. (22) in the revised manuscript due to a change in equation numbering) yields the correct Gibbs free energy at constant P for the α phase, and $\Delta G(\text{Pb} \rightarrow \text{Sn})$ reflects the correct free energy difference between the α phases.

However, in the original manuscript and SI, we described the alchemical path as “more relevant” for computing free energy differences. In light of the reviewer’s comment, we acknowledge that this description does not provide an accurate or fair comparison between the two approaches, since one could also perform Frenkel–Ladd calculations for the β phase to obtain the $\beta \rightarrow \alpha$ free energy difference. Accordingly, we have removed the discussion of relevancy in the following:

- **Main text Sec. II C, page 8**

The Frenkel–Ladd path is simulated ~~using a fixed cell (NVT)~~in the fixed cell (NVT) of the respective \$\alpha\$ phase, whereas the alchemical path is simulated in the ~~more-relevant~~ NPT ensemble, in which phase transformations can occur.

- **SI Sec. III, page 4**

~~Since our definition of phase in the main text is based on the highest free energy barrier—the transition between perovskite (P) and non-perovskite (N) structures—the alchemical path simulated in the NPT ensemble, which accounts for the lower free energy perovskite phase (α or β) at each temperature, aligns better with our goal of estimating free energy differences.~~

We also note that the primary comparison here concerns the computational efficiency of the alchemical pathway, particularly in terms of reduced dissipated energy and faster convergence, enabled by larger phase space overlap. Therefore, the main conclusion—showing matching free energies at high temperatures with improved efficiency—still demonstrate the usefulness of this approach.